# A thiol-bound drug reservoir enhances APR-246-induced mutant p53 tumor cell death

Sophia Ceder[1] ⓘ, Sofi E Eriksson[1] ⓘ, Emarndeena H Cheteh[1] ⓘ, Swati Dawar[2] ⓘ,
Mariana Corrales Benitez[2], Vladimir J N Bykov[1] ⓘ, Kenji M Fujihara[2,3] ⓘ, Mélodie Grandin[4,5] ⓘ,
Xiaodun Li[6] ⓘ, Susanne Ramm[7], Corina Behrenbruch[3,4], Kaylene J Simpson[7] ⓘ,
Frédéric Hollande[4,5] ⓘ, Lars Abrahmsen[8], Nicholas J Clemons[2,3] ⓘ & Klas G Wiman[1,*] ⓘ

## Abstract

The tumor suppressor gene *TP53* is the most frequently mutated gene in cancer. The compound APR-246 (PRIMA-1Met/Eprene-tapopt) is converted to methylene quinuclidinone (MQ) that targets mutant p53 protein and perturbs cellular antioxidant balance. APR-246 is currently tested in a phase III clinical trial in myelodysplastic syndrome (MDS). By *in vitro*, *ex vivo*, and *in vivo* models, we show that combined treatment with APR-246 and inhibitors of efflux pump MRP1/ABCC1 results in synergistic tumor cell death, which is more pronounced in *TP53* mutant cells. This is associated with altered cellular thiol status and increased intracellular glutathione-conjugated MQ (GS-MQ). Due to the reversibility of MQ conjugation, GS-MQ forms an intracellular drug reservoir that increases availability of MQ for targeting mutant p53. Our study shows that redox homeostasis is a critical determinant of the response to mutant p53-targeted cancer therapy.

**Keywords** APR-246; Eprenetapopt; glutathione; MRP1; p53
**Subject Categories** Cancer; Pharmacology & Drug Discovery

## Introduction

The tumor suppressor p53 is a transcription factor that regulates multiple cellular processes including cell cycle arrest, metabolism, redox homeostasis, and cell death in response to cellular stress (Mello & Attardi, 2018). *TP53* is the most frequently mutated gene in cancer (Kandoth *et al*, 2013), and *TP53* mutation is associated with chromosomal instability (Donehower *et al*, 2019) and poor

prognosis according to some studies (Olivier *et al*, 2006; Bally *et al*, 2014). The majority of *TP53* mutations are missense mutations which disrupt p53-dependent transcription (Soussi & Wiman, 2015). Mutant p53 may also in some settings have gain-of-function (GOF) activities that promote tumor growth (Muller & Vousden, 2014).

The mutant p53-targeting compounds PRIMA-1 and APR-246 (PRIMA-1Met/Eprenetapopt) are converted to the active product methylene quinuclidinone (MQ), a Michael acceptor that reacts with thiols (Lambert *et al*, 2009). MQ has been shown to bind preferentially to Cys124 and Cys277 (Wassman *et al*, 2013; Zhang *et al*, 2018) of the 10 cysteine (Cys) residues in the DNA-binding core domain of p53 (Meplan *et al*, 2000). Treatment of mutant p53-expressing cells with PRIMA-1 or APR-246 results in mutant p53 reactivation and cell death (Bykov *et al*, 2002b). A clinical phase I/II study has shown that APR-246 is safe and well tolerated (Lehmann *et al*, 2012). APR-246 is currently undergoing a phase III clinical trial in myelodysplastic syndrome (MDS) and several phase II clinical trials (https://clinicaltrials.gov).

We and others have previously shown that APR-246 (through MQ) exhibits pro-oxidant activities through depletion of glutathione (GSH) and inhibition of the GSH and thioredoxin (TXN) antioxidant systems (Peng *et al*, 2013; Tessoulin *et al*, 2014; Mohell *et al*, 2015; Haffo *et al*, 2018; Hang *et al*, 2018; Mlakar *et al*, 2019). GSH, present at steady state millimolar concentrations, is crucial for maintaining cellular thiol redox homeostasis. Cystine (CySS)/glutamate system xCT (SLC7A11) imports one of the building blocks of GSH, and tumor cells with high levels of GSH and high xCT expression are more resistant to APR-246 (Mohell *et al*, 2015; Liu *et al*, 2017; Mlakar *et al*, 2019). Therefore, the redox effects of APR-246 are important for its anti-tumor activity. Since p53 has a redox regulatory role (Sablina *et al*, 2005; Kruiswijk *et al*, 2015) and its activity is dependent on the redox environment (Hainaut & Milner, 1993; Ueno *et al*, 1999), separating APR-246 anti-tumor activities mediated

1   Department of Oncology-Pathology, Karolinska Institutet, Stockholm, Sweden
2   Peter MacCallum Cancer Centre, Melbourne, Vic., Australia
3   Sir Peter MacCallum Department of Oncology, The University of Melbourne, Parkville, Vic., Australia
4   Department of Clinical Pathology, The University of Melbourne, Melbourne, Vic., Australia
5   Victorian Comprehensive Cancer Centre, University of Melbourne Centre for Cancer Research, Melbourne, Vic., Australia
6   MRC Cancer Unit, University of Cambridge, Cambridge, UK
7   Peter MacCallum Cancer Centre, Victorian Centre for Functional Genomics, Melbourne, Vic., Australia
8   Aprea Therapeutics AB, Solna, Sweden
    *Corresponding author. Tel: +46 73 986 6586; E-mail: Klas.Wiman@ki.se

by mutant p53 from those mediated by altered redox balance is complex (Eriksson et al, 2019).

The efflux transporter multidrug resistance-associated protein 1 (MRP1/ABCC1) mediates drug resistance (Walsh et al, 2010; Bagnoli et al, 2013; Cole, 2014) as it exports both endogenous and xenobiotic molecules conjugated to GSH (Cole, 2014; Whitt et al, 2016). MRP1 also has a redox regulatory role by controlling GSH/GSSG homeostasis through the export of GSH and the oxidized form GSSG (Minich et al, 2006; Cole, 2014). Analysis of the NCI database demonstrated that the expression of MRP family proteins can contribute to PRIMA-1 resistance (Bykov et al, 2002a).

The association of MRP expression with PRIMA-1 resistance, MRP1's role in redox regulation and export of GSH-conjugated drugs, and the fact that MQ conjugates to GSH led us to hypothesize that inhibition of MRP1 in mutant TP53-carrying tumor cells enhances sensitivity to APR-246. We show here that inhibition of MRP1 or xCT/SLC7A11 synergizes with APR-246 to induce cancer cell death by increasing intracellular drug availability and by altering cellular redox status. We also demonstrate that GS-MQ adduct formation is reversible, indicating that GS-MQ could serve as an intracellular drug reservoir to promote mutant p53 reactivation.

# Results

### Inhibition of efflux pump MRP1 synergizes with APR-246

ABCC1/MRP1 is one of the genes with the highest correlation to PRIMA-1 resistance in ovarian cancer cells according to data available through the Cancer Dependency Map (DepMap) (Fig EV1A) (Basu et al, 2013; Seashore-Ludlow et al, 2015; Rees et al, 2016) consistent with our NCI database analysis (Bykov et al, 2002a). Indeed, combination of APR-246 with the MRP1 inhibitor MK-571 resulted in enhanced growth suppression in TP53 mutant TOV-112D and OVCAR-3 ovarian cancer cell lines and HCT116 colorectal cancer cells with WT or R248W mutant TP53 (Fig 1A), as well as several esophageal cancer cell lines (Fig EV1B) and cancer cell lines of other origins and different TP53 status (Appendix Fig S1A). In contrast, normal human dermal fibroblasts (HDFs) were relatively insensitive to single and combination treatments (Fig 1B). Clustering analysis divided the cell lines into high and low sensitivity to APR-246 groups where most lines in the high sensitivity group carry mutant TP53 (Appendix Fig S1B). IC$_{50}$ values of APR-246 were significantly lowered upon incubation with MK-571 (Fig 1C and Appendix Table S1). We observed synergy in essentially all cancer cell lines (Fig 1D and Appendix Fig S1C), but cells with mutant TP53 showed stronger synergy than WT TP53 cells (Fig 1D). This difference is statistically significant (P = 0.02, unpaired t-test) if FLO-1 (Fig EV1B) with a TP53 mutation at Cys277, one of the cysteines targeted by MQ (Zhang et al, 2018), is excluded. In our relatively small panel of tested cell lines, MRP1 protein expression did not show any obvious association with TP53 gene status, APR-246 IC$_{50}$, or δ-score (Appendix Fig S1D). However, as MRP1 efflux activity is GSH- and ATP-dependent (Hooijberg et al, 2000), MRP1 protein expression may not necessarily reflect its efflux pump activity. Synergistic growth suppression by APR-246 and MK-571 combination treatment was also shown in OVCAR-3 cells by flow cytometry (sub-G1 DNA content) (Fig 1E and Appendix Fig S1F)

and real-time cell confluence using Incucyte® (Fig 1F and Appendix Fig S7O–Q).

Combining APR-246 with Reversan, another MRP1 inhibitor, also led to synergistic growth suppression and cell death in HCT116 TP53 R248W and WT cells (Fig 1G and Appendix Fig S1G). Partial silencing of MRP1 using four different siRNAs separately (Fig 1H, and Appendix Fig S1H and I) increased sensitivity to APR-246 in both HCT116 WT and R248W TP53 cells (Fig 1I, and Appendix Fig S1J and K) whereas overexpression of MRP1 (Fig 1J and Appendix Fig S1L) resulted in resistance to APR-246 (Fig 1K and Appendix Fig S1M).

In summary, blocking MRP1 transporter activity using two chemically distinct inhibitors, MK-571 or Reversan, or siRNA-mediated downregulation of MRP1, significantly potentiates APR-246-induced cancer cell death.

### MK-571 enhances the anti-tumor activity of APR-246 in vivo and ex vivo

The in vivo anti-tumor effect of APR-246 has previously been confirmed in mice with Eso26 esophageal cancer cell xenografts (TP53 R248W). While APR-246 at 100 mg/kg strongly inhibited tumor growth and prolonged survival (Liu et al, 2015), we observed no effect of 50 mg/kg APR-246 on tumor volume (Fig 2A and Appendix Fig S2A and B) and survival (Fig 2B). However, combination treatment with MK-571 significantly enhanced APR-246 anti-tumor activity (Fig 2A, and Appendix Fig S2A and B) and survival (Fig 2B) in MRP1-expressing Eso26 xenografts (Appendix Fig S2C), consistent with the in vitro data on esophageal cell lines (Fig EV1B). At an early time point (5 days of treatment), before any significant changes in tumor volumes, we observed a decrease in proliferation marker Ki67 staining in all drug treatment groups (Fig 2C), while there was no induction of apoptosis marker cleaved caspase 3 at this timepoint (Fig 2D). At late time points (> 22 days after treatment initiation), we observed an increased cleaved caspase 3 staining in both APR-246 and the combination treatment group (Fig 2D and E). p53 staining confirmed high expression of mutant p53 (Fig 2E).

We next assessed the effect of APR-246 and MK-571 treatment in patient-derived organoids (PDO) (Appendix Table S2). PDOs recapitulate tumor heterogeneity and are useful pre-clinical tools for evaluation of precision therapy (Li et al, 2018). For the first time, we demonstrate APR-246 anti-tumor efficacy in PDO cultures derived from TP53 missense mutant colorectal cancer (colo-PDO) (Fig 2F) and esophageal adenocarcinoma (eso-PDO) (Appendix Fig S2D), as assessed by the ATP-based CTG assay. IC$_{50}$ values for APR-246 were significantly lowered by co-treatment with MK-571 in tested PDOs (Fig 2G). Combination treatment of APR-246 and MK-571 resulted in synergistic growth suppression in colo-PDOs according to the metabolic CTG assay (Fig 2F). Combination treatment in eso-PDOs resulted in a synergistic response at specific concentrations (Appendix Fig S2E and F). Prior to CTG signal quantification, colo-PDOs were stained with propidium iodide (PI) and Hoechst and imaged (Fig EV2A and B) to access cell death and enable image analysis-based synergy determination (Appendix Fig S2G). The image analysis showed that organoid area and PI staining were relatively unchanged upon APR-246 treatment in colo-PDOs (Fig 2H), despite the measured changes in growth suppression/metabolic activity (Fig 2F). However, organoid area decreased and PI intensity increased dramatically upon addition of MK-571 (Fig 2H). Synergy

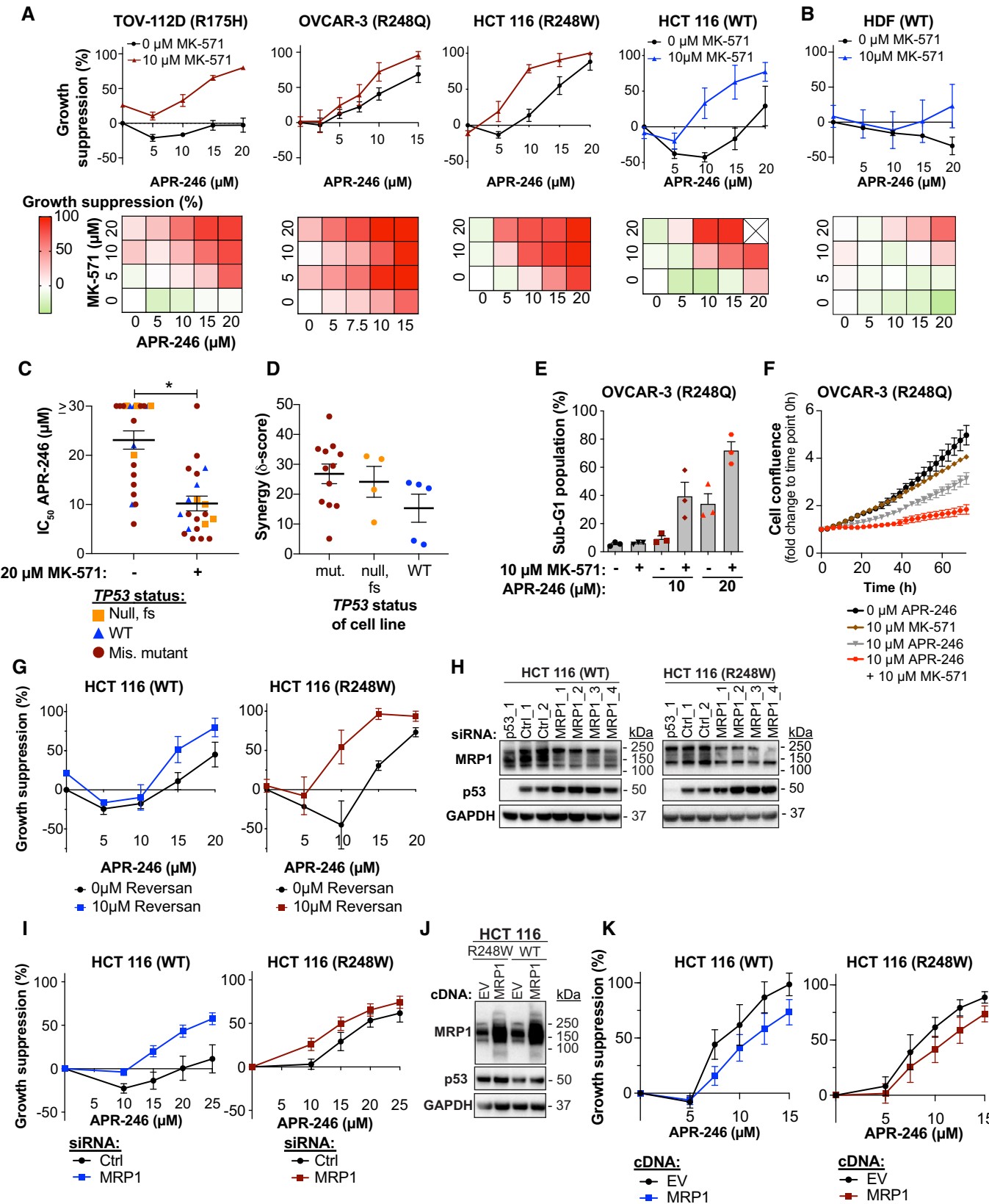

Figure 1.

**Figure 1. Inhibition of efflux pump MRP1 synergizes with APR-246.**

A   Growth suppression in cancer cells after 72 h of treatment with APR-246 +/− MRP1 inhibitor MK-571 as shown by the WST-1 assay ($n \geq 3$, $n$ shown in Appendix Table S1). Cross in heatmap indicates no data available.

B   Growth suppression in human dermal fibroblasts (HDF) by WST-1 assay after 72 h with APR-246 +/− MK-571 ($n = 3$).

C   Mean $IC_{50}$ values (μM) for APR-246 +/− 20 μM MK-571 of 21 cell lines. *$P < 0.0001$, Wilcoxon test. Each dot represents one cell line. Mean $IC_{50}$ values and $n$ shown in Appendix Table S1.

D   ZIP synergy scores of most synergistic area sub-grouped according to *TP53* gene status for 21 cell lines. Each dot represents one cell line. Scores and $n$ are shown in Appendix Table S1.

E   Sub-G1 DNA content of OVCAR-3 cells (R248Q) after APR-246 +/− MK-571 treatment as assessed by propidium iodide staining at 48 h ($n = 3$).

F   OVCAR-3 cell confluency by IncuCyte® during 72 h of treatment with APR-246 +/− MK-571 ($n = 3$).

G   Growth suppression in HCT116 WT ($n = 4$) and R248W cells ($n = 3$) after 72-h treatment with APR-246 +/− MRP1 inhibitor Reversan, as shown by WST-1.

H   Western blot analysis of MRP1 (Cell Signaling), p53 (DO-1), and GAPDH of HCT116 WT and R248W cells 48 h after transfection with negative control siRNAs or siRNAs targeting MRP1 or p53.

I   Growth suppression (WST-1 assay) of HCT116 WT and R248W cells transfected with MRP1 siRNA ($n \geq 5$ and $n \geq 2$, respectively) after 48 h APR-246 treatment. Indicated values are average of four individual siRNAs targeting MRP1 and two individual negative control siRNAs. Data and $n$ of individual siRNAs are shown in Appendix Fig S1J; data are also part of Fig 7B.

J   Western blot analysis of MRP1 (Cell Signaling), p53 (DO-1), and GAPDH of HCT116 WT and R248W cells 72 h after transfection with empty vector (EV) and MRP1 expression vector.

K   Growth suppression (WST-1 assay) of HCT116 WT and R248W cells transfected with empty vector (EV) and MRP1 expression vector after 72 h APR-246 treatment ($n = 4$). Data are also part of Appendix Fig S7E.

Data information: *TP53* status is shown for each cell line. Data are represented as mean ± SEM. See also Fig EV1, Appendix Fig S1 and Appendix Table S1.
Source data are available online for this figure.

---

analysis by the ZIP model revealed synergistic growth suppression in both eso-PDOs and colo-PDOs according to the ATP-based CTG assay and organoid area assessment, and also synergistic cell death as assessed by PI intensity. This was evident both from analysis of the maximum synergy (Fig 2I) and the average synergy over the concentration range tested (Appendix Fig S2H).

In conclusion, MK-571 potentiates APR-246 efficacy *in vitro* in cultured cancer cell lines, *in vivo* in mouse xenografts, and *ex vivo* in patient-derived organoids.

## Inhibition of MRP1 efflux pump activity increases $^{14}$C-APR-246/MQ accumulation in cancer cells

Radiolabeled $^{14}$C-APR-246 in which $^{14}$C is retained in the active conversion product MQ was used to study uptake and accumulation of radioactivity in various cell lines (Fig 3A). A linear relationship between $^{14}$C-APR-246 concentration and $^{14}$C accumulation was apparent after 6 h of drug exposure (Appendix Fig S3A). Treatment with $^{14}$C-APR-246 in combination with MK-571 caused a significant increase in $^{14}$C accumulation over APR-246 treatment alone as assessed in 11 different cell lines (Figs 3B and EV3, Appendix Fig S3B and Appendix Table S3). MK-571 blocked MRP1 pump activity already after 6-h treatment according to accumulation of MRP1 substrate doxorubicin (Appendix Fig S3C; Deeley & Cole, 2006; Zhang *et al*, 2016). The $^{14}$C signal determined after 24 h was normalized to protein concentration to take possible APR-246-mediated cell death into account. We observed no obvious association of $^{14}$C signal with *TP53* status. Reversan enhanced $^{14}$C accumulation to a similar extent as MK-571 in the isogenic HCT116 cell lines with either WT or R248W mutant *TP53* (Fig 3C and Appendix Fig S3D). Likewise, downregulation of MRP1 using four different siRNAs separately (Fig 1H) increased cellular $^{14}$C content independently of *TP53* status in HCT116 cells (Fig 3D and Appendix Table S4).

Our results demonstrate that $^{14}$C-APR-246 accumulates inside cells upon MRP1 inhibition or siRNA knockdown, suggesting that either APR-246, MQ, MQ conjugates, or all three are exported via MRP1.

## MRP1 inhibition increases GS-MQ content, forming a reservoir of MQ for interaction with other targets

Liquid chromatography mass spectrometry (LC-MS) was used to assess drug content in OVCAR-3 cells after 24-h incubation with APR-246. This treatment had only minor effects on cell viability (Appendix Fig S4A and B). Whereas intracellular APR-246 content after co-treatment with MK-571 remained unchanged (Fig 4A), levels of GS-MQ increased (Fig 4B). The stability of GS-MQ was assessed *in vitro* at pH 7.4 by incubating GS-MQ with the thiol-containing compound N-acetyl cysteine (NAC) (Fig EV4). The concentration of GS-MQ remained essentially constant over 24 h in buffer at room temperature (Fig 4C). However, incubation with equimolar concentrations of NAC resulted in a marked decrease in the amounts of GS-MQ within 30 min and a concomitant increase in NAC-MQ (Fig 4D), indicating that MQ is in rapid flux between a conjugated form and a free form, that becomes apparent in the presence of a suitable donor, e.g., free thiol group(s). Therefore, GS-MQ formed within cells might be viewed as a reservoir of MQ from which MQ can be released and bind to other cellular targets.

These data suggest that the inhibition of MRP1 does not lead to accumulation of prodrug APR-246 but results in retention of GS-MQ within the cell. This increases the cellular pool of active product MQ, which can bind to various intracellular thiols.

## APR-246 sensitivity is dependent on the presence of mutant p53, cellular thiol status, and drug accumulation

Cells expressing mutant p53 were more sensitive to APR-246 than cells lacking p53 (Appendix Table S1, Figs 5A and EV5A), in agreement with our previous data (Bykov *et al*, 2005). Mutant p53-expressing cells in all three isogenic cell systems tested (HCT116, H1299, and Saos-2) showed a more pronounced synergistic growth

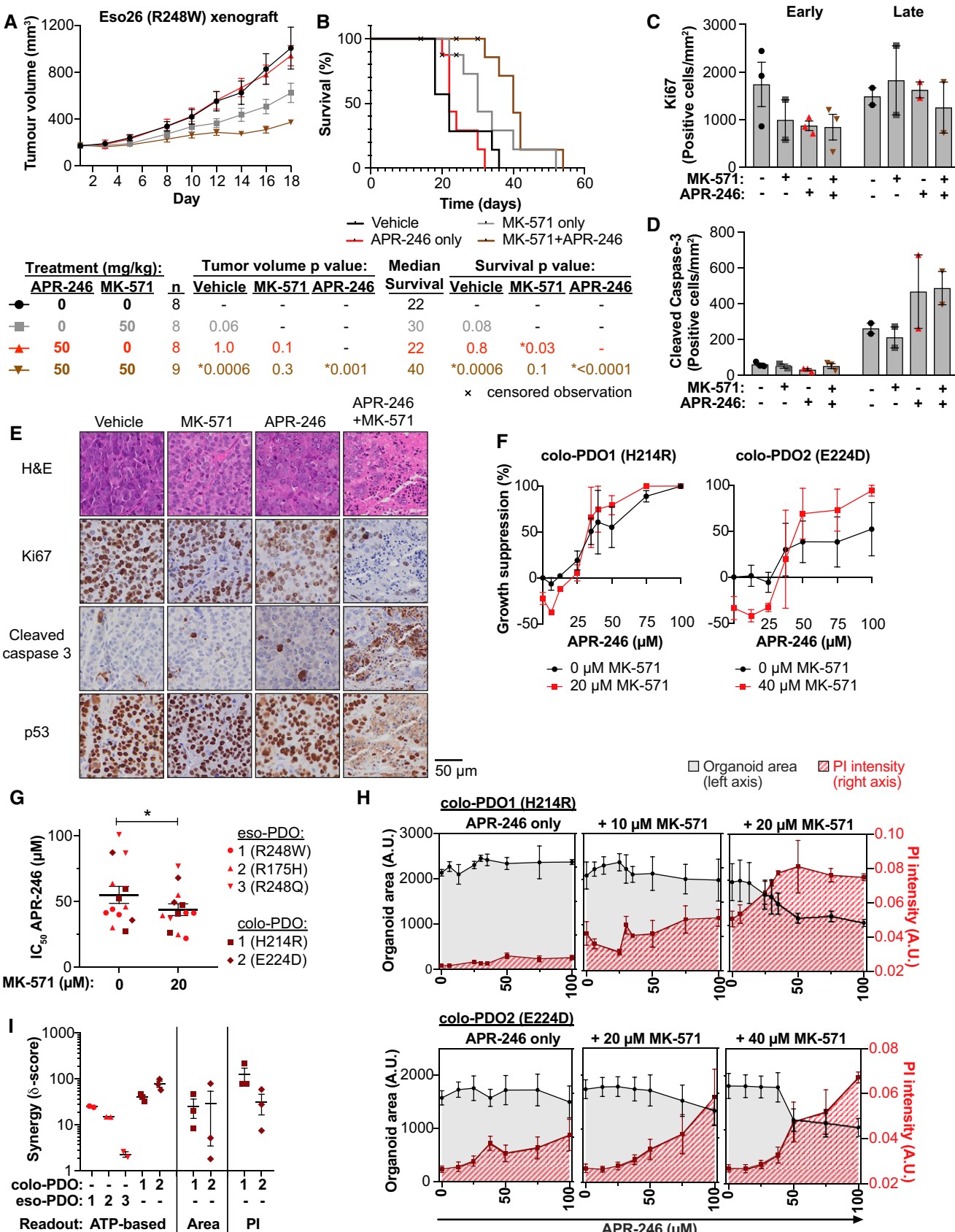

Figure 2.

**Figure 2.** MK-571 enhances the anti-tumor activity of APR-246 *in vivo* and *ex vivo*.

A   Growth of Eso26 xenografts (R248W mutant *TP53*) in mice treated with vehicle, APR-246 (50 mg/kg), MK-571 (50 mg/kg), or APR-246 and MK-571 for 16 days. *P* values determined by one-way ANOVA with Tukey's correction at day 18, *n* is indicated in the figure.

B   Kaplan–Meier plot of time to reach 1,400 mm³ tumor volume in mice treated with vehicle, APR-246 (50 mg/kg), MK-571 (50 mg/kg), or APR-246 and MK-571 for 16 days. Mice that were culled at ethical endpoints prior to reaching a tumor volume of > 1,400 mm³ are marked as censored observations. *P* values were determined by Log-rank (Mantel–Cox) test between different treatment groups. *n* is indicated in the figure.

C   Quantification of Ki67 immunohistochemistry staining of Eso26 xenograft tumors at an early timepoint (5 days) or post-treatment/late timepoint (> 22 days) after treatment initiation with APR-246 (50 mg/kg) +/− MK-571 (50 mg/kg). Dots indicate individual tumors.

D   Quantification of cleaved caspase 3 immunohistochemistry staining of Eso26 xenograft tumors in same tumors as shown in Fig 2C. Dots indicate individual tumors.

E   Representative images of hematoxylin/eosin (H&E) staining and immunostaining of Ki67, cleaved caspase 3 and p53, post-treatment/late timepoint (> 22 days after treatment initiation) with APR-246 (50 mg/kg) +/− MK-571 (50 mg/kg).

F   Growth suppression determined by the ATP-based CTG assay in colorectal cancer patient-derived organoids (colo-PDO) after treatment with APR-246 +/− MK-571 (*n* = 3).

G   IC50 values (μM) APR-246 +/− 20 μM MK-571 in indicated PDOs (colo-PDO or esophageal adenocarcinoma patient-derived PDO [eso-PDO]). *P* = 0.04, paired *t*-test (*n* = 3 except colo-PDO2 and eso-PDO3 where *n* = 2).

H   Organoid area (in black, left *y*-axis) and PI intensity (in red, right *y*-axis) of colo-PDO1 and colo-PDO2 as determined by image analysis 72 h after treatment with APR-246 +/− MK-571 (*n* = 3).

I   Highest synergy score according to the ZIP model based on growth suppression as shown by CTG assay (ATP-based) or image analysis (Area and PI) in eso-PDOs and colo-PDOs. Scores are shown on a log scale. Score above 0 indicates synergy. Each dot indicates one individual experiment.

Data information: *TP53* status is indicated. Data are represented as mean ± SEM. See also Fig EV2, Appendix Fig S2 and Appendix Table S2.
Source data are available online for this figure.

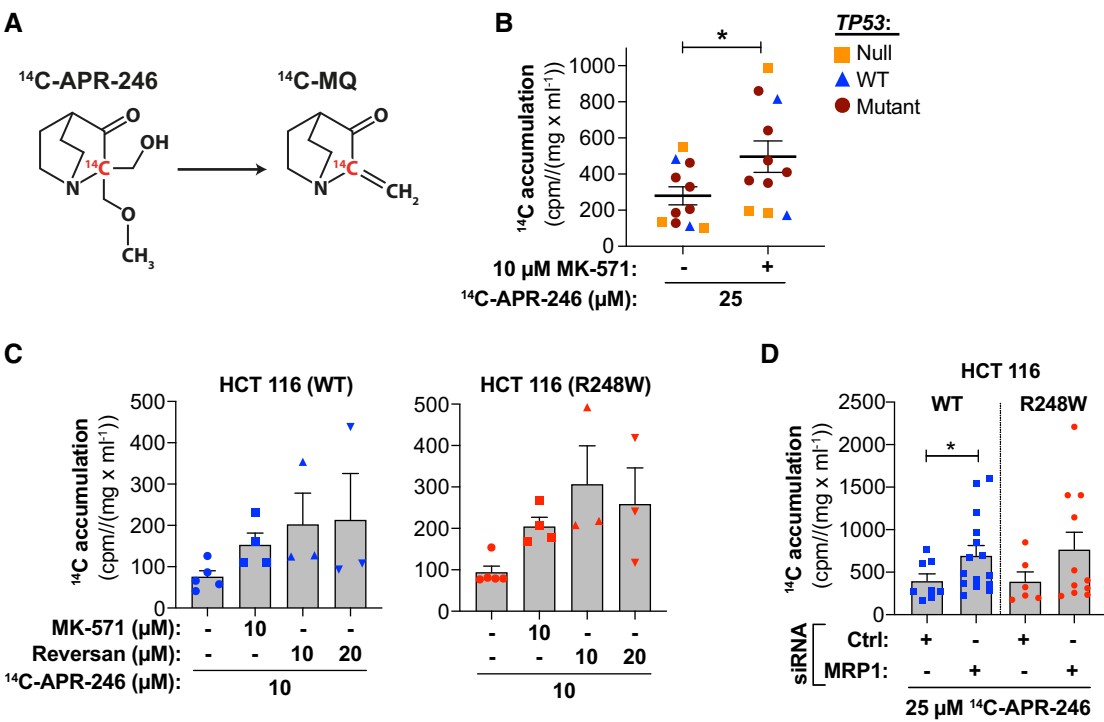

**Figure 3.** Inhibition of MRP1 efflux pump activity increases ¹⁴C-APR-246/MQ accumulation in cancer cells.

A   Chemical structure of ¹⁴C-APR-246 and its active product MQ.

B   ¹⁴C accumulation (cpm/mg/ml) in 11 cell lines after 24 h ¹⁴C-APR-246 +/− MK-571 treatment (*n* ≥ 3). *P* = 0.0006, paired *t*-test. ¹⁴C-accumulation and *n* of individual cell lines as shown in Appendix Table S3.

C   ¹⁴C accumulation (cpm/mg/ml) in HCT116 (WT and R248W) cells after 24 h + treatment with ¹⁴C-APR-246 +/− MK-571 or Reversan (*n* ≥ 3, each dot indicates one individual experiment).

D   ¹⁴C accumulation (cpm/mg/ml) in HCT116 WT and R248W cells (*n* ≥ 3 and ≥ 2) transfected with MRP1 siRNAs after 24 h of ¹⁴C-APR-246 treatment. Indicated values represent four individual MRP1 siRNAs and two individual negative control (Ctrl) siRNAs, where each dot is an individual siRNA. Mean of ¹⁴C-accumulation after treatment and *n* with individual siRNA sequences is shown in Appendix Table S4. *P* = 0.04 and for R248W *P* = 0.06, Wilcoxon matched-pairs signed rank test.

Data information: *TP53* status has been indicated. Data are represented as mean ± SEM. See also Fig EV3, Appendix Fig S3 and Appendix Tables S3 and S4.

   

suppression compared with p53 null and WT cells (Appendix Table S1). Liu *et al* (2017) showed that tumor cells with *TP53* mutation can exhibit reduced *de novo* synthesis of GSH due to mutant p53 gain-of-function (GOF) activities that increase sensitivity to APR-246. Indeed, GSH and GSSG are the strongest predictive metabolites for response to PRIMA-1, based on examination of 225 metabolites available in the DepMap portal (Li *et al*, 2019) (Figs 5B and EV5B). However, our data suggest that the association between *TP53* status and GSH concentration is cell context-dependent (Fig 5C and D, and Appendix Fig S5A–C). H1299 lung cancer cells with exogenous R175H mutant *TP53* had a lower basal concentration of total GSH + GSSG as compared to parental p53 null H1299 cells or doxycycline-treated H1299 tet-off R175H cells (Fig 5C, and Appendix Fig S5A and B). In contrast, among the isogenic HCT116 colorectal cancer cells, p53 null cells had a lower level of total GSH + GSSG compared with both WT and mutant *TP53* HCT116 cells (Fig 5D). In Saos-2 cells, there was no detectable difference in total GSH + GSSG levels between parental p53 null and untreated and doxycycline-treated tet-off R273H mutant p53-expressing cells (Appendix Fig S5C). Thus, total GSH + GSSG level is not the sole factor determining sensitivity to APR-246. Interestingly, assessment

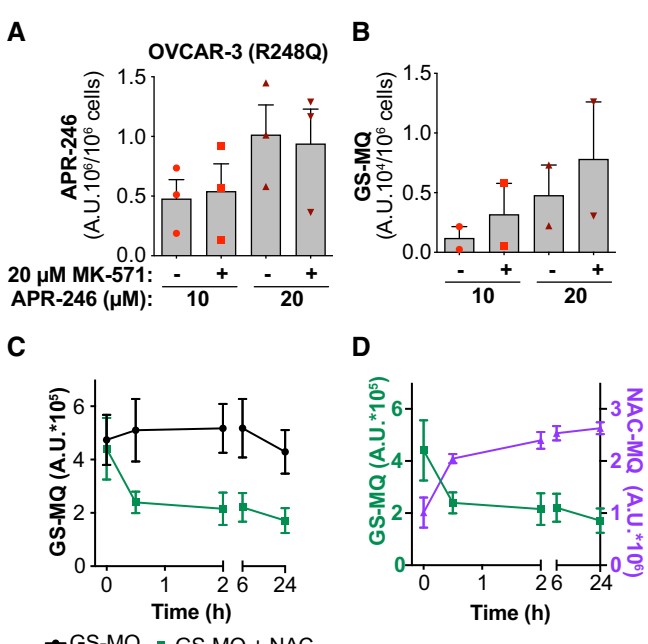

**Figure 4. MRP1 inhibition increases GS-MQ content, forming a reservoir of MQ for interaction with other targets.**

A  APR-246 content in OVCAR-3 cells (TP53 R248Q) 24 h after treatment with APR-246 +/− MK-571 (*n* = 3).

B  Glutathione-conjugated MQ (GS-MQ) in OVCAR-3 cells after 24-h treatment with APR-246 +/− MK-571 (*n* = 2).

C  Amount of GS-MQ at indicated timepoints after incubation of GS-MQ +/− NAC (*n* = 2). See Fig EV4 for chemical reaction.

D  Amount of GS-MQ by LC-MS (green line, left axis) and NAC-MQ (purple line, right axis) over time after incubation of GS-MQ with NAC (*n* = 2). Values on *y*-axes are not comparable because GS-MQ and NAC-MQ have different response signals on MS.

Data information: Indicated values are assessed by LC-MS. Data are represented as mean ± SEM. See also Appendix Fig S4.

using ThiolTracker™, a dye that reacts with thiols in intact cells, showed that the observed difference in total GSH + GSSG (Fig 5C and D) is not reflected in the global thiol status of the cell (Fig 5E).

Accumulation of [14]C after treatment with [14]C-APR-246 showed a significant correlation with $IC_{50}$ values for APR-246 (Fig 5F). Hence, cells that accumulate more APR-246/MQ are also more sensitive, irrespective of *TP53* status. Nonetheless, analysis of The Cancer Genome Atlas (TCGA) PanCancer Atlas (Hoadley *et al*, 2018) from cBioportal (Cerami *et al*, 2012; Gao *et al*, 2013) revealed that in some tumors types, *ABCC1* (MRP1) mRNA is higher in patients with putative driver *TP53* missense mutations compared with wild-type *TP53* (Fig 5G). This difference was evident in lung cancer (lung adenocarcinoma and lung squamous cell carcinoma), colon adenocarcinoma, and to some extent esophageal carcinoma (Appendix Fig S5D). This suggests that *TP53* mutant tumor cells of these cancer types may have a better capacity to export drugs. However, as mentioned above, our data show that mutant p53-expressing cells are more sensitive to APR-246. To further examine the role of mutant p53 in APR-246 accumulation, mutant p53 expression in tet-off R175H cells was repressed using doxycycline and accumulation of [14]C-APR-246/MQ was assessed (Fig 5H and Appendix Fig S5E). Inhibition of mutant p53 expression did not affect [14]C-APR-246/MQ accumulation but still decreased APR-246-induced growth suppression (Fig 5I and Appendix Fig S5F). However, the combination treatment was nonetheless effective in these cells (Appendix Fig S5G). Similarly, siRNA knockdown of R248W mutant p53 in HCT116 cells (Appendix Fig S5H) did not affect total GSH + GSSG level (Fig EV5C) or [14]C content after treatment with [14]C-APR-246 (Fig EV5D and Appendix Table S4), but nonetheless reduced APR-246-induced growth suppression (Fig EV5E).

Thus, sensitivity to APR-246 treatment is dependent on mutant *TP53* status, GSH content, and drug accumulation, but neither of these factors alone can fully explain sensitivity to APR-246.

### MRP1 inhibitor MK-571 shifts cellular thiol pools, further potentiating APR-246 efficacy

Given the reversible nature of MQ conjugation, the potential impact of the intracellular thiol milieu on APR-246-induced cell death in the presence of MK-571 was investigated. We detected a decrease in total GSH + GSSG after MK-571 treatment in several cell lines (Figs 6A and EV6 and Appendix Fig S6A), in agreement with previous studies (Cullen *et al*, 2001; Hirrlinger *et al*, 2002; Minich *et al*, 2006). This decrease in total GSH + GSSG after MK-571 treatment may explain the single agent activity of MK-571 in reducing tumor volume (Fig 2A).

LC-MS analysis of OVCAR-3 cells revealed a minor decrease in oxidized GSH (GSSG) 24 h after APR-246 treatment (Fig 6B), while GSH was unchanged at this time point (Fig 6C). The combination treatment with APR-246 and MK-571 caused a marked decrease in both GSH and GSSG levels (Fig 6B and C). Cysteine (Cys), a key building block for GSH synthesis, is imported in the oxidized form CySS by antiporter xCT in exchange for glutamate (Glu) (Nunes & Serpa, 2018). Expression levels of xCT varied in our panel of cell lines (Appendix Fig S6B). OVCAR-3 cells that showed low levels of xCT upregulated antiporter expression upon MRP1 inhibition by MK-571 (Fig 6D and Appendix Fig S6C). Similar results were observed in HCT116 WT and R248W cells (Fig 6E and Appendix Fig S6D) as well as other cancer cell lines (Appendix Fig S6E–G). This is most likely a compensatory upregulation in response to the decreased

concentration of GSH (Fig 6A and C). Imported CySS may be efficiently reduced to Cys by enzymes present in the intracellular compartment (Espinosa & Arner, 2019). The elevated expression of xCT was accompanied by an increase in the intracellular CySS and Cys content, consistent with increased xCT antiporter activity after MK-571 treatment (Fig 6F and G). Interestingly, H1299 $TP53^{-/-}$ and HCT116 $TP53^{-/-}$ cells exhibited higher basal expression of xCT compared with their isogenic counterparts carrying mutant or WT $TP53$ and did not show an adaptive increase in xCT expression upon MK-571 treatment (Appendix Fig S6G). Decreased intracellular Cys levels were observed upon combination treatment with APR-246 and MK-571, suggesting that increased intracellular Cys is consumed, modified, or generated to a lesser extent during the combination treatment (Fig 6G). Cells exposed to APR-246 and MK-571 had a higher level of xCT (Fig 6D and E, and

Appendix Fig S6E and F) and lower level of intracellular Glu (Appendix Fig S6H), also consistent with increased xCT activity.

In conclusion, MK-571 treatment alters the intracellular thiol status and depletes total GSH + GSSG. Due to the subsequent increase in xCT expression, the thiol pool shifts toward elevated intracellular CySS and Cys levels (Fig 6F and G). Hence, MK-571 potentiates the cell death-inducing activity of APR-246 through a dual mechanism: inhibition of drug export and perturbation of intracellular thiol pools.

## GSH and Cys availability determines APR-246/MQ accumulation and sensitivity to APR-246

Partial knockdown of antiporter xCT after $^{14}$C-APR-246 treatment led to a roughly twofold increase in $^{14}$C accumulation in WT and

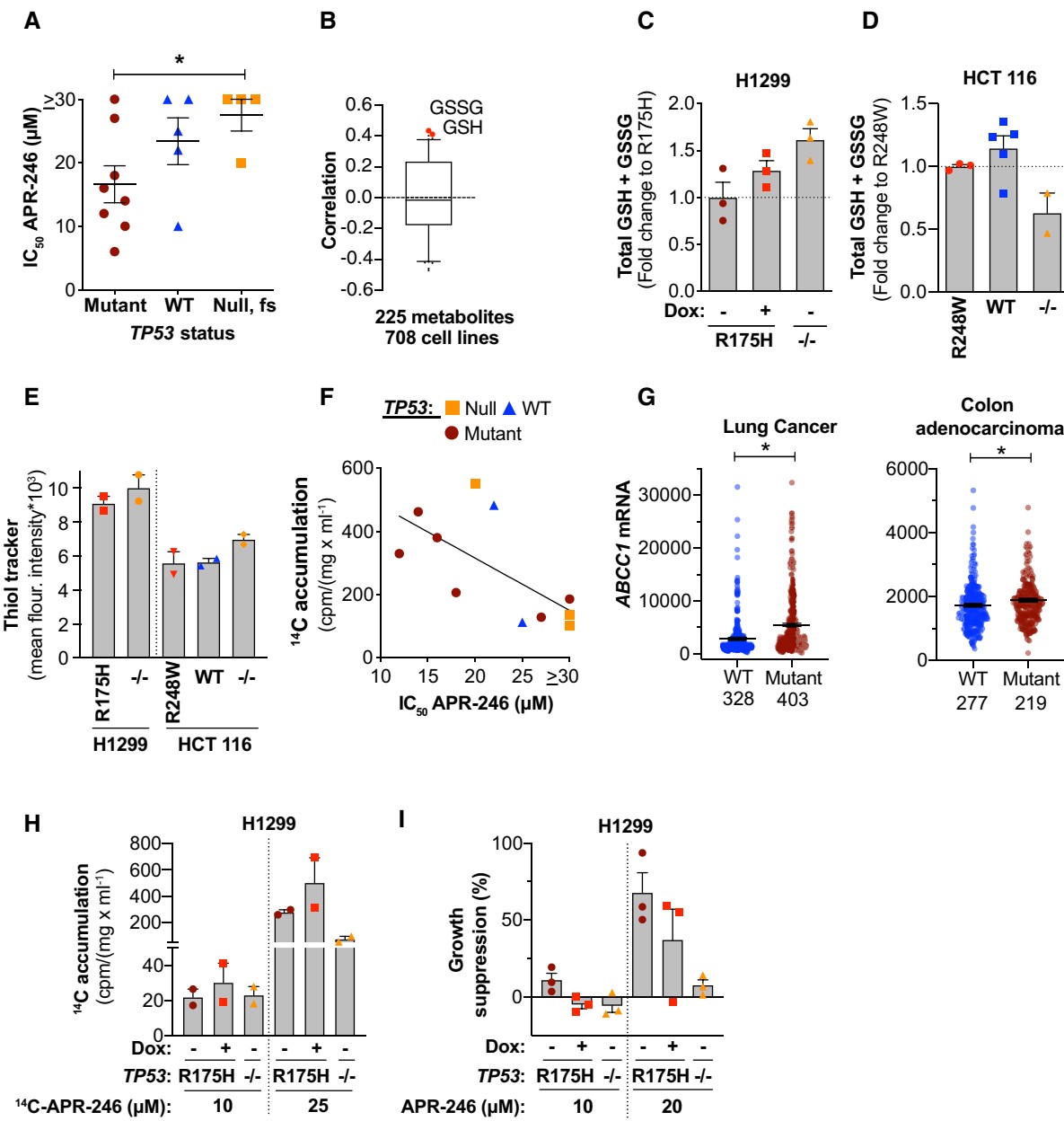

**Figure 5.**

◀ **Figure 5. APR-246 sensitivity is dependent on the presence of mutant p53, cellular thiol status and drug accumulation.**

A  Mean $IC_{50}$ (μM) values of APR-246 in 17 different cell lines ($n \geq 3$) as shown by WST-1. *$P = 0.03$, Mann–Whitney test. Mean $IC_{50}$ (μM) values for APR-246 treatment are also shown in Fig 1C. Values and $n$ are listed in Appendix Table S1.

B  Box-and-whisker plot of Pearson correlations between PRIMA-1 area-under-the-curve (AUC) and the levels of 225 different metabolites from the DepMap portal. Central band indicates median, boxes indicate $25^{th}$ and $75^{th}$ percentile, and whiskers show $2.5^{th}$ and $97.5^{th}$ percentile outlier metabolites. High GSH and GSSG (in red) correlate with low PRIMA-1 sensitivity.

C  Total intracellular glutathione (GSH + GSSG) as shown by glutathione reductase (GR) re-cycling assay in H1299 R175H (tet-off) (+/− doxycycline) and H1299 −/− cells 24 h after seeding ($n = 3$). Indicated values are fold change compared with mutant *TP53* cells.

D  Total intracellular glutathione (GSH + GSSG) as shown by GR re-cycling assay in untreated HCT116 (R248W, WT, and −/−) cells 48 h after seeding ($n = 3, 5$, or 2, respectively). Indicated values are fold change to mutant *TP53* cells. Part of this data is shown in Fig EV6.

E  Thiol tracker staining by flow cytometry 48 h after seeding of the indicated untreated cells ($n = 2$).

F  Correlation between $^{14}$C accumulation (cpm/mg/ml) after treatment with 25 μM $^{14}$C-APR-246 and $IC_{50}$ (μM) values for APR-246 in 11 cell lines ($n \geq 3$) ($r = −0.66$, $P = 0.03$, Pearson correlation). Each dot represents one cell line, $^{14}$C accumulation data are shown in Fig 3B and values and $n$ for individual cell lines are listed in Appendix Table S1 and S3.

G  *ABCC1* mRNA Expression, RSEM (Batch normalized from Illumina HiSeq_RNASeqV2) in lung cancer (luad & lusc studies, *$P < 0.0001$) and colon adenocarcinoma (coad study, *$P = 0.005$) grouped into having no alterations or putative driver missense mutations in *TP53*. Statistical analysis by Mann–Whitney test, $n$ is indicated in the figure.

H  $^{14}$C accumulation (cpm/mg/ml) in H1299 R175H cells (+/− doxycycline) and H1299 −/− cells after 24-h treatment with $^{14}$C-APR-246 ($n = 2$).

I  Growth suppression in H1299 R175H cells (+/− doxycycline) and H1299 −/− cells treated for 72 h with APR-246 as shown by WST-1 assay ($n = 3$).

Data information: *TP53* status is indicated for each cell line. Data are represented as mean ± SEM. See also Fig EV5 and Appendix Fig S5.

Source data are available online for this figure.

---

R248W mutant *TP53* HCT116 cells (Fig 7A, Appendix Fig S7A and Appendix Table S4). Simultaneous xCT knockdown and MRP1 inhibition by MK-571 caused a three- and fourfold increase in $^{14}$C accumulation (Fig 7A and Appendix Table S4) and increased APR-246-induced growth suppression (Appendix Fig S7B).

Silencing of either xCT or MRP1 resulted in a similar potentiation of APR-246 activity in HCT116 *TP53* WT cells (Fig 7B, Appendix Fig S7C, left panel and Appendix Fig S1J and K). Interestingly, xCT silencing had a more potent effect than MRP1 knockdown on APR-246-induced growth suppression in HCT116 cells with mutant *TP53* (Fig 7B, Appendix Fig S7C, right panel, and Appendix Fig S1J and K). This is consistent with results showing that xCT can promote resistance to APR-246 in mutant *TP53*-harboring tumor cells (Appendix Fig S7D and E) (Liu *et al*, 2017). Accordingly, we also assessed the relative levels of xCT in our cell line panel (Appendix Fig S7F). Higher levels of xCT were indeed associated with higher APR-246 $IC_{50}$ values, i.e., higher resistance (Appendix Fig S7G), in agreement with data from the DepMap portal with 708 cancer cell lines (Ghandi *et al*, 2019) (Appendix Fig S7H and I) and a previous study (Liu *et al*, 2017). High xCT is also associated with higher APR-246 $IC_{50}$ values in combination treatment with MK-571 when cell lines are grouped according to xCT levels (Appendix Fig S7J). Furthermore, downregulation of xCT sensitized to APR-246 more than downregulation of MRP1 in mutant TP53 cells (Fig 7B right panel), even though the cells accumulated drug to a similar extent (Fig 7A right panel and Appendix Table S4).

Partial knockdown of MRP1 or xCT at 24 h did not significantly affect the total GSH + GSSG content in cells (Fig 7C and Appendix Fig S7K). Thus, the observed increase in $^{14}$C accumulation after siRNA knockdown of xCT (Fig 7A) could not be explained by changes in total GSH + GSSG levels (Fig 7C), but may instead be related to other thiols. The xCT inhibitor sulfasalazine (SSZ) at 200 μM decreased total GSH + GSSG to similar extent as 10 μM MK-571 after 3 h (Appendix Fig S7L) and 24 h of treatment (Fig 7D). Nonetheless, SSZ treatment resulted in significantly higher intracellular $^{14}$C accumulation in HCT116 cells (Fig 7E and

Appendix Fig S7M). Similar data were obtained with H1299 cells (Appendix Fig S7N). We observed pronounced growth suppression by combined treatment with APR-246 and SSZ as compared to APR-246 alone in both HCT116 WT and R248W cells, as assessed by total cellular protein in cell culture wells (Fig 7F). These findings show that APR-246 drug dynamics inside cells is not only dependent on GSH levels but also on CyS/CySS content. Real-time confluence analysis in OVCAR-3 cells confirmed that both MK-571 and SSZ potentiate APR-246-induced growth suppression (Fig 7G and H and Appendix Fig S7O–Q).

In summary, MK-571 and SSZ synergize with APR-246 by affecting both GSH/GSSG and Cys/CySS availability, which in turn dictates APR-246/MQ accumulation and APR-246-induced growth suppression.

## Discussion

MQ, the active conversion product of APR-246, promotes stability of the folded p53 core domain (Lambert *et al*, 2009; Zhang *et al*, 2018) and compromises major antioxidant defense systems (Peng *et al*, 2013; Tessoulin *et al*, 2014; Mohell *et al*, 2015; Haffo *et al*, 2018; Hang *et al*, 2018; Mlakar *et al*, 2019). Previous studies have indicated that the dependence of APR-246-induced growth suppression and cell death on mutant p53 varies with cellular background (Tessoulin *et al*, 2014; Grellety *et al*, 2015; Mohell *et al*, 2015; Sobhani *et al*, 2015; Ali *et al*, 2016; Liu *et al*, 2017). Two aspects of the mechanism of MQ make it challenging to determine the relative importance of mutant p53 reactivation and induction of oxidative conditions for APR-246-induced cell death; first, mutant p53 restoration may synergize with oxidative injuries in promoting tumor cell death and, second, the thiol-binding properties of MQ underlies both effects. Therefore, the use of, for example, NAC to prevent oxidative conditions in APR-246-treated cells will also result in the capture of MQ, and as a consequence, less MQ will be available to target mutant p53.

The combination of APR-246 with MRP1 inhibitors or MRP1 siRNA knockdown produced strong synergistic cell death in our

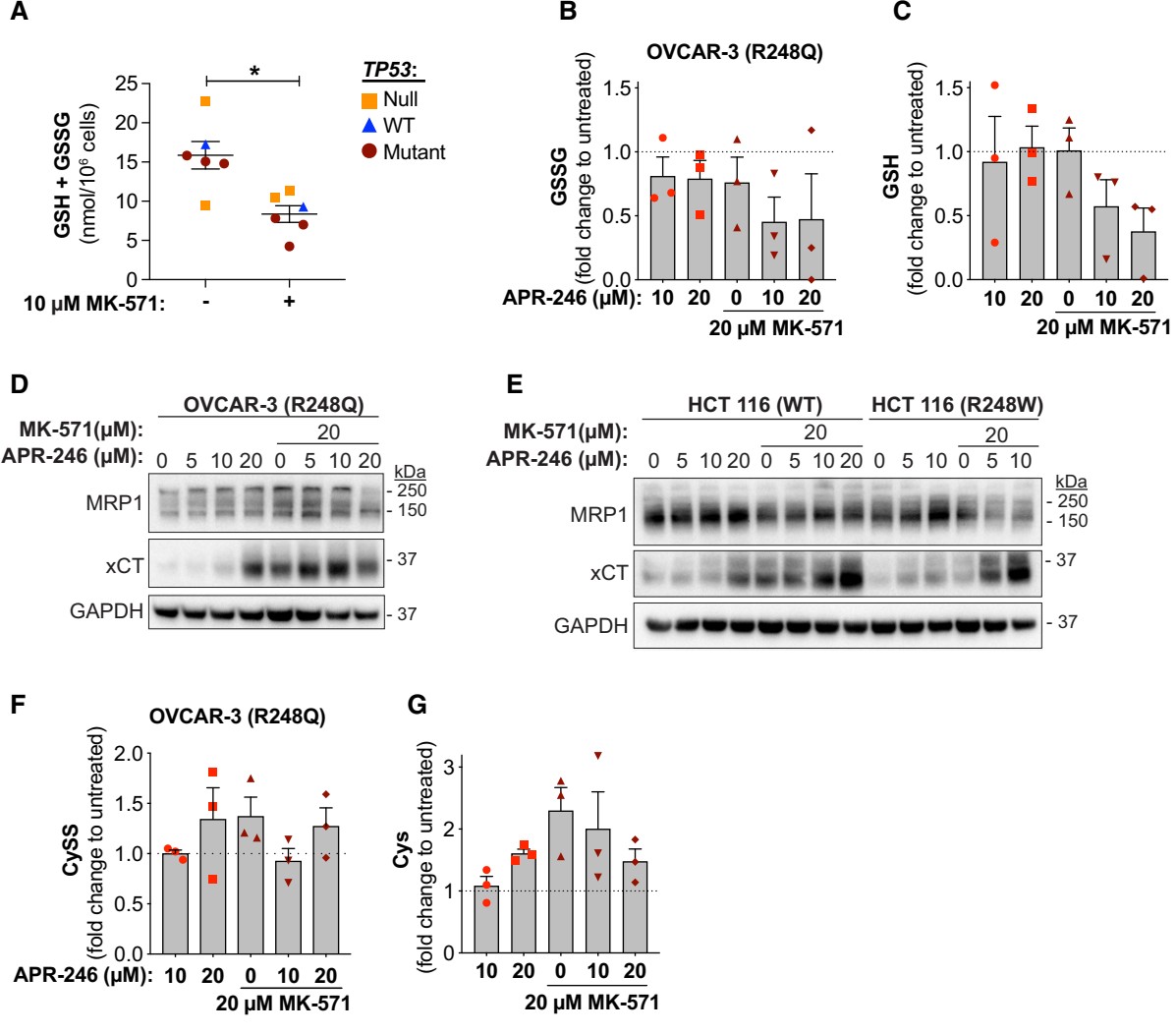

**Figure 6. MRP1 inhibitor MK-571 shifts cellular thiol pools, further potentiating APR-246 efficacy.**

A  Total intracellular glutathione (GSH + GSSG per $10^6$ cells), as determined by GR re-cycling assay after 24-h MK-571 treatment in six different cell lines ($n \geq 3$, except HCT116 WT $n = 2$). *$P$ = 0.014, Paired *t*-test. Each dot represents one cell line, see Fig EV6 for individual cell lines and $n$.

B  Intracellular oxidized glutathione (GSSG per $10^6$ cells) as shown by LC-MS in OVCAR-3 (R248Q TP53) cells at 24-h treatment with APR-246 +/− MK-571 ($n = 3$).

C  Intracellular reduced glutathione (GSH per $10^6$ cells) as shown by LC-MS in OVCAR-3 cells at 24-h treatment with APR-246 +/− MK-571 ($n = 3$).

D  Western blot analysis of MRP1 (Cell Signaling), xCT, and GAPDH of OVCAR-3 cells treated with APR-246 +/− MK-571 for 24 h.

E  Western blot analysis of MRP1 (Cell Signaling), xCT, and GAPDH of HCT116 WT and R248W cells treated with APR-246 +/− MK-571 for 24 h.

F  Intracellular cystine (CySS per $10^6$ cells) as shown by LC-MS in OVCAR-3 cells at 24 h treatment with APR-246 +/− MK-571 ($n = 3$).

G  Intracellular cysteine (Cys per $10^6$ cells) as shown by LC-MS in OVCAR-3 cells at 24 h treatment with APR-246 +/− MK-571 ($n = 3$).

Data information: *TP53* status is indicated. Data are represented as mean ± SEM. See also Fig EV6 and Appendix Fig S6.
Source data are available online for this figure.

panel of tumor cells but not in normal HDFs (Fig 1). Importantly, we showed that MK-571 treatment could lower the threshold of APR-246 anti-tumor activity in esophageal cancer xenografts in mice and patient-derived organoids (PDOs) (Fig 2). We found that $^{14}$C accumulation after $^{14}$C-APR-246 treatment was associated with APR-246-induced growth suppression (Fig 5F) and inhibition of MRP1 by inhibitors or siRNA caused a substantial increase in intracellular amount of $^{14}$C (Fig 3). LC-MS analysis showed that this was a result of increased accumulation of GS-MQ, whereas no significant accumulation of APR-246 was observed (Fig 4A and B). Thus, our data

suggest that the synergy observed upon combination treatment with APR-246 and MRP1 efflux inhibitor or siRNA knockdown is at least in part due to increased intracellular accumulation of GS-MQ and/or other MQ adducts.

The preferential reactivity of Michael acceptor MQ with soft nucleophiles, for instance thiols, derives from the α,β-unsaturated carbonyl (Jackson *et al*, 2017). Our time course experiment showed that the Michael reaction was rapidly reversible (Fig 4C and D) which provides a molecular explanation for the previously observed reversible inhibition of thioredoxin and glutaredoxin (Haffo *et al*,

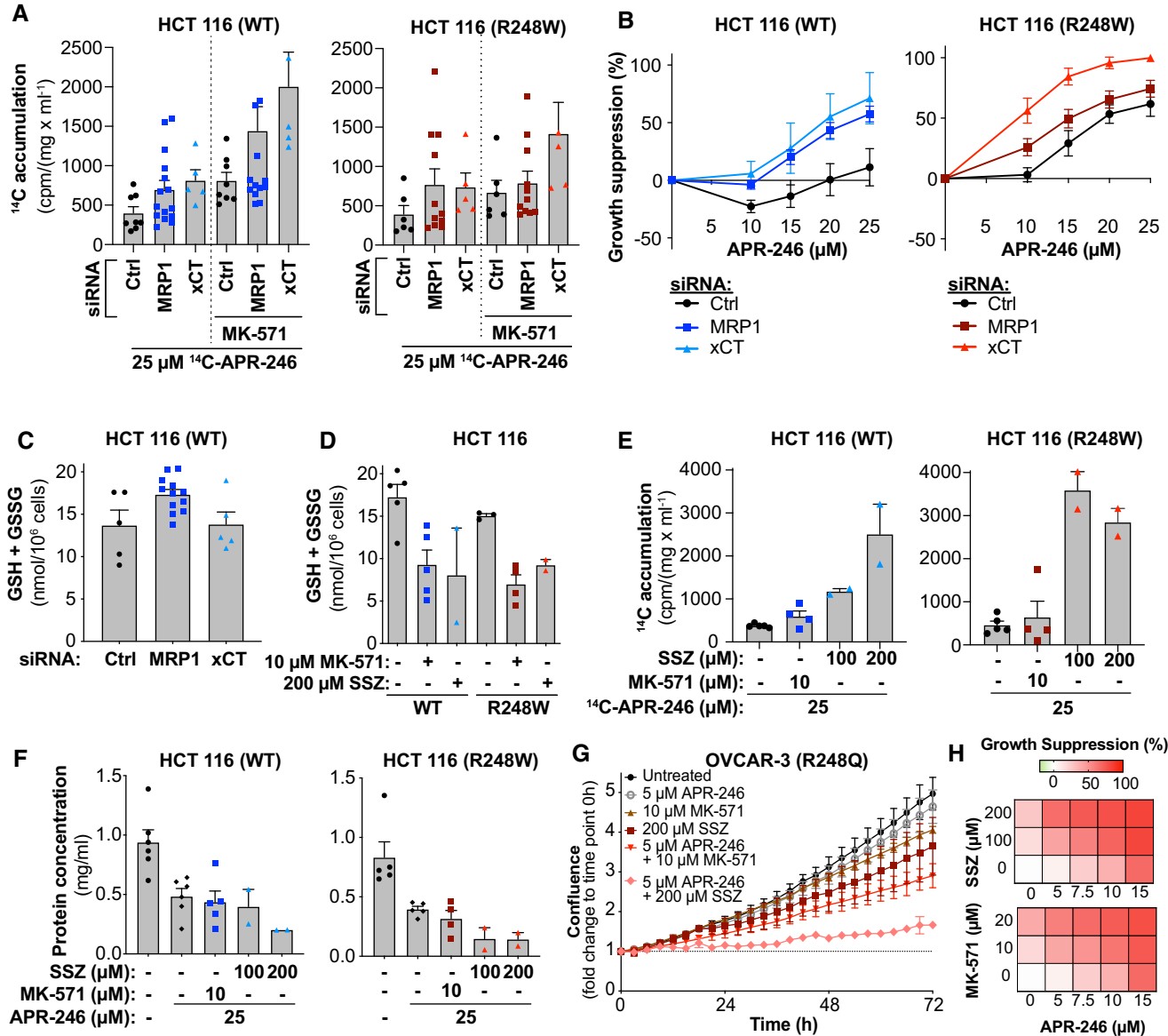

**Figure 7. GSH and Cys availability determines APR-246/MQ accumulation and sensitivity to APR-246.**

A $^{14}$C accumulation (cpm/mg/ml) in HCT116 WT and R248W mutant *TP53* cells transfected with MRP1 or xCT siRNAs following 24 h of $^{14}$C -APR-246 treatment +/− MK-571. Indicated values are mean values from four MRP1 siRNAs ($n \geq 3$), three xCT siRNAs ($n = 1$–2) and two control siRNAs ($n \geq 3$). Mean $^{14}$C accumulation after treatment and $n$ of individual siRNAs is shown in Appendix Table S4. Data for control and MRP1 siRNAs after $^{14}$C -APR-246 treatment is also shown in Fig 3D.

B Growth suppression in HCT116 WT and R248W cells transfected with MRP1 or xCT siRNAs after 48 h of APR-246 treatment as shown by WST-1 assay. Indicated values are mean values of four MRP1 siRNAs ($n \geq 3$), three xCT siRNAs ($n = 1$–2), and two control siRNAs ($n \geq 3$). Data for control and MRP1 siRNAs after APR-246 single treatment are also included in Fig 1I. Values and $n$ for individual siRNAs shown in Appendix Fig S7C.

C Total intracellular glutathione (GSH + GSSG per $10^6$ cells) in HCT116 WT cells as shown by GR re-cycling assay 48 h post-transfection of siRNA. Indicated values are means from four MRP1 siRNAs ($n = 3$), two xCT siRNAs ($n = 2$–3), and two control siRNA ($n = 2$–3). Values and $n$ for individual siRNAs shown in Appendix Fig S7K.

D Total intracellular glutathione (GSH + GSSG per $10^6$ cells) in HCT116 WT and R248W cells measured by GR re-cycling assay after 24 h of MK-571 or sulfasalazine (SSZ) treatment ($n \geq 3$, except SSZ where $n = 2$, $n$ indicated by dots).

E $^{14}$C accumulation (cpm/mg/ml) in HCT116 WT and R248W cells at 24-h treatment with $^{14}$C -APR-246, combined with MK-571 or sulfasalazine (SSZ), ($n \geq 4$, except SSZ where $n = 2$, $n$ indicated by dots). Data for $^{14}$C -APR-246 +/− MK-571 is also shown in Fig 3C and Appendix Fig S3D.

F Protein concentration of HCT116 WT and R248W assessed by DC™ Protein assay at 24 h treatment with APR-246 +/− MK-571 or sulfasalazine (SSZ), or untreated ($n \geq 4$, except SSZ where $n = 2$, $n$ indicated by dots).

G OVCAR-3 (R248Q) confluence by IncuCyte® during 72-h treatment with APR-246 +/− MK-571 or sulfasalazine (SSZ) ($n = 3$). Part of the data are shown in Fig 1F.

H Growth suppression in OVCAR-3 cells as determined by IncuCyte® during 72 h treatment with APR-246 +/− MK-571 or sulfasalazine (SSZ) depicted as heatmaps ($n = 3$).

Data information: Data are represented as mean ± SEM. See also Appendix Fig S7 and Appendix Table S4.

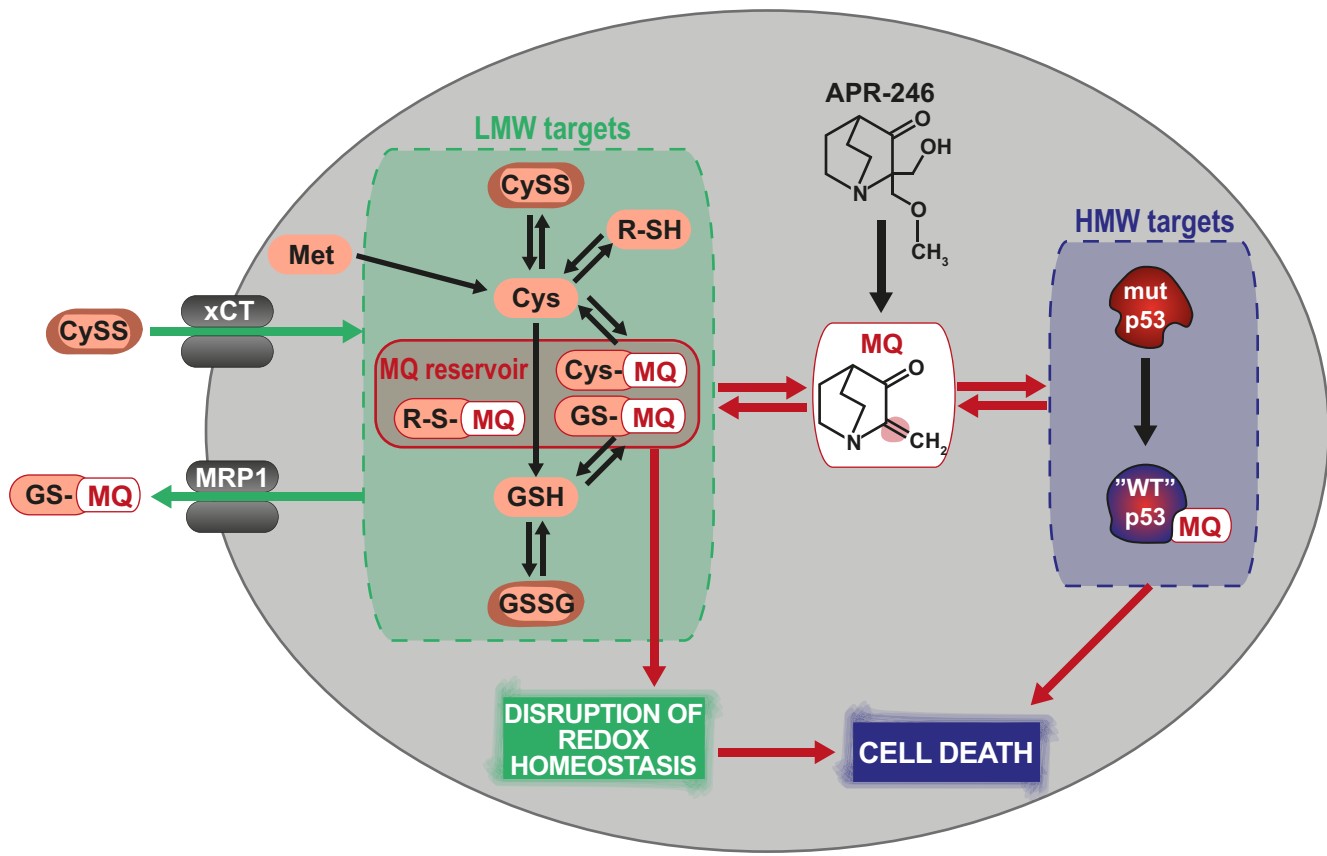

**Figure 8. Thiol-bound MQ provides a drug reservoir that increases availability of MQ for targeting mutant p53.**

MQ can bind both high molecular weight (HMW) targets such as mutant p53 and low molecular weight (LMW) targets, e.g., GSH and Cysteine. MQ conjugation to GSH (GS-MQ) is reversible, indicating that MQ can transfer between multiple HMW and LMW cellular targets. Thus, GS-MQ serves as an intracellular drug reservoir. MRP1 regulates GSH and oxidized GSH (GSSG) content and exports GS-MQ. Therefore, MRP1 inhibition potentiates the effect of APR-246/MQ through a dual mechanism: (1) inhibition of GS-MQ efflux and (2) disruption of the intracellular thiol balance sensitizing cells to oxidative stress. We postulate that an MQ reservoir is similarly formed by conjugation to other thiol (R-SH) targets, e.g., Cys. Furthermore, xCT and MRP1 collectively regulate LMW target availability and thus the intracellular reservoir of MQ, thereby governing sensitivity to APR-246.

2018). Hence, entrapping GS-MQ is equivalent to creating an intracellular MQ pool. Furthermore, as shown by LC-MS, GS-MQ conjugates seemed stable over 24-h incubation at room temperature, while the reversibility became apparent when the competing thiol compound NAC was added (Fig 4C and D). This indicates that the steady state concentration of free MQ in APR-246-treated cells is very low and that MQ rapidly travels between various cellular thiol targets, including known targets such as mutant p53 (Lambert *et al*, 2009), thioredoxin reductase (Peng *et al*, 2013), thioredoxin, glutaredoxin, and ribonucleotide reductase (Haffo *et al*, 2018). We hypothesize that a transient stabilization of mutant p53 by MQ binding is sufficient to induce p53 tetramerization on DNA, driving p53-dependent transcription. This is consistent with data showing that brief exposure of R175H mutant p53 to $Zn^{2+}$ is sufficient to induce mutant p53 reactivation and apoptosis (Yu *et al*, 2018). The rapid reversibility and hence inherent instability of the MQ conjugates make detection technically challenging, which may explain why we were not able to detect Cys-MQ conjugates. Thus, our data suggest that the trapping of GS-MQ within tumor cells by inhibiting MRP1-mediated drug efflux can potentiate the effect of APR-246/MQ by

forming an intracellular MQ reservoir, from which MQ can be released to target for instance thiols in mutant p53.

Treatment with MK-571 caused a decrease in intracellular total GSH + GSSG concentration in a majority of the tested cell lines (Fig 6A–C), in agreement with previous reports (Cullen *et al*, 2001; Hirrlinger *et al*, 2002; Minich *et al*, 2006). Since APR-246 treatment increases total protein glutathionylation (Haffo *et al*, 2018), accumulation of MQ upon combination treatment with MRP1 inhibitors might further enhance intracellular glutathionylation, leading to a decrease in total GSH + GSSG. Hence, MRP1 inhibition alone or in combination with APR-246 treatment causes major changes in the cellular GSH + GSSG content.

It is conceivable that a higher intracellular concentration of MQ in the context of diminished GSH levels will allow more extensive modification of key mutant p53 cysteines or other protein thiols, and thus more efficient mutant p53 activation and subsequent tumor cell killing. Consequently, the lowering of total intracellular GSH + GSSG level upon treatment with MK-571 probably contributes to the increased potency of APR-246. The synergy between APR-246 and MK-571 was more prominent in cells carrying mutant

*TP53* (Fig 1D), but there was no obvious correlation between [14]C accumulation and *TP53* status (Fig 5F and H, and EV5D). Thus, transient downregulation of R175H mutant p53 did not affect [14]C accumulation (Fig 5H), nor did it abrogate the synergistic cell death induced by combination treatment with APR-246 and MK-571 (Appendix Fig S5G). This shows that additional factors besides total MQ accumulation and *TP53* status determine the extent of synergy.

MRP1 has been reported to mediate efflux of the cytotoxic lipid peroxidation product 4-hydroxynonenal (4-HNE) as a GSH conjugate (Renes *et al*, 2000; Cole, 2014). Considering that APR-246 can induce lipid peroxidation in tumor cells (Liu *et al*, 2017), it seems likely that increased intracellular accumulation of toxic byproducts of lipid peroxidation following MRP1 inhibition contributes to the observed synergy. Thus, both lipid peroxidation and perturbation of redox homeostasis might contribute to the observed oxidative DNA damage after APR-246 treatment (Liu *et al*, 2017; Hang *et al*, 2018; Ogiwara *et al*, 2019).

The xCT antiporter regulates *de novo* GSH synthesis by importing the precursor CySS, which can subsequently be reduced to Cys. APR-246 also synergizes with the xCT inhibitor sulfasalazine (SSZ) *in vitro* (Fig 7G) and *in vivo* (Liu *et al*, 2017). In the present study, we found that MK-571 treatment leads to increased xCT protein levels in tumor cells carrying WT or mutant *TP53* (Fig 6D and E, and Appendix Fig S6E–G). High amounts of xCT in OVCAR-3 cells were accompanied by elevated intracellular Cys levels (Fig 6G). Elevated Cys levels are presumably a result of increased activity of xCT, although a contribution of for instance the transsulfuration pathway cannot be excluded (McBean, 2017). Increased xCT protein levels after MK-571 treatment could be interpreted as a response to shifts in the intracellular Cys/CySS and GSH/GSSG ratio, which involves induction of *SLC7A11* (xCT) via transactivation by ATF4 and NRF2 (Ishii & Mann, 2014; Yu & Long, 2016). Interestingly, several studies have shown that APR-246 treatment leads to an increased xCT protein level in tumor cells (Ali *et al*, 2016; Liu *et al*, 2017; Lisek *et al*, 2018; Synnott *et al*, 2018), possibly due to its inhibition of mutant p53-NRF2 complexing and/or its pro-oxidant activities (Walerych *et al*, 2016; Eriksson *et al*, 2019). However, despite the compensatory upregulation of xCT at 24 h, the growth suppression induced by combination treatment with MK-571 and APR-246 was substantial (Fig 1).

siRNA knockdown of xCT in HCT116 *TP53* WT or R248W cells increased both [14]C levels and sensitivity to APR-246, although the total GSH + GSSG level was not altered (Fig 7A–C). xCT controls Cys/CySS status and GSH synthesis, but also hydrogen sulfide signaling, taurine production, and protein synthesis (Carter & Morton, 2016). Hence, this is a complex system and xCT and the Cys/CySS status might influence tumor cell viability and sensitivity to APR-246 through several mechanism. This notion is further supported by our data on [14]C accumulation upon combination treatment with [14]C-APR-246, MK-571, and siRNA against MRP1 or xCT. The robust increase in [14]C accumulation in xCT-deficient cells treated with APR-246 plus MK-571 (Fig 7A) suggests that MRP1 and xCT have complementary activities. Inhibition of MRP1 and xCT depleted GSH to similar extent, yet xCT inhibition led to a higher accumulation of [14]C, suggesting that cysteine availability plays an important role for export of MQ. Furthermore, it has been suggested that MK-571 treatment inhibits secretion of Cys via MRP1 (Olm

*et al*, 2009). Thus, we speculate that Cys-MQ is also entrapped by MRP1 inhibition. However, Cys and consequently GSH are both depleted upon xCT inhibition with SSZ, which leads to decreased export of MQ and increased binding of MQ to other thiols, including protein thiols.

In summary, we have demonstrated synergy between APR-246 and MK-571 in *in vitro* cultured cancer cells, *in vivo* in mice, and *ex vivo* in esophageal and colorectal cancer PDOs. The PDO model is considered to have strong predictive value for clinical efficacy of novel cancer therapy (Li *et al*, 2018). We demonstrate that sensitivity to APR-246 is governed by both MRP1 efflux pump activity and cellular low molecular weight (LMW) thiol status, i.e., levels of GSH/GSSG and Cys/CySS. Thus, the strong synergistic effect of APR-246 and MK-571 combination treatment is most likely due to the co-targeting of high molecular weight (HMW) thiols such as mutant p53 (leading to p53 core domain refolding), cellular redox homeostasis (depletion of GSH + GSSG, accumulation of lipid peroxides, and Cys/CySS imbalance), and drug export (accumulation of an intracellular MQ pool). The reversibility of MQ adduct formation is a key aspect of the mechanism of action of APR-246 (Fig 8) which may also in part account for the benign safety profile reported from clinical studies of APR-246 (Lehmann *et al*, 2012).

# Materials and methods

### Cell lines and culturing conditions

Detailed information about source, origin, *TP53* status, and culturing conditions for all cell lines is provided in Appendix Table S5. Experiments were performed 1 week after thawing of cells up to 8 weeks in culture at the latest unless otherwise specified. Cells were frozen as early as possible after thawing. Saos-2 R273H and H1299 R175H cells carry exogenous tetracycline-regulated mutant p53 (Tet-off) (Bykov *et al*, 2002b). H1299 R175H cells were treated with 2 µg/ml doxycycline for 2 days and then continuously with 0.5 µg/ml doxycycline for at least 1 week to turn off expression of mutant p53. Saos-2 R273H cells were treated continuously with 5 µg/ml doxycycline for at least 1 week to turn off expression of mutant p53. Doxycycline was present during experiments. HCT116 *TP53* null and R248W cells were generated by targeted disruption of endogenous WT *TP53* or knockin of R248W-mutant *TP53* (Bunz *et al*, 1998). KADA is a newly established melanoma cell line (Ligtenberg *et al*, 2018). Cells were counted by Countess™ II, Thermo Fisher Scientific, USA.

### Patient-derived organoids (PDO) culturing conditions

Experiments conformed to the principles set out in the WMA Declaration of Helsinki and the Department of Health and Human Services Belmont Report. Experiments involving esophageal cancer patient-derived organoids (eso-PDO) were registered (UKCRNID 8880) and approved by the Institutional Ethics Committee (REC07/H0305/52 and 10/H0305/1), all subjects gave individual informed consent. The colorectal cancer patient-derived organoids (colo-PDO) were grown from samples collected under human ethic agreement HREC/15/PMCC/112, Peter MacCallum Cancer Centre Human Research Ethics Committee, Melbourne, Australia, and signed

informed consents were obtained from all patients prior to samples acquisition. Detailed information regarding culturing media of the PDOs is included in Appendix Table S2. Derivation and culture of eso-PDO (Li *et al*, 2018) colo-PDO (Paquet-Fifield *et al*, 2018) is previously described. For culturing colo-PDOs, size and confluency was determined using bright-field microscopy. For passaging of organoids, media was removed and 500 μl ice-cold phosphate-buffered saline (PBS) was added to dissolve the Matrigel and collect the organoids. The organoids were centrifuged at 350 *g* for 5 min, and supernatant was discarded. The pellet, either after collecting or dissociating, was resuspended in 10 ml cold PBS and centrifuged at 350 *g* for 5 min for a second wash. The supernatant was discarded, and the pellet was resuspended in Matrigel (Corning® Matrigel® Growth Factor Reduced phenol red-free, BDAA356231, Bio-Strategy, Melbourne, VIC, Australia) (diluted with basal culture medium [BCM] to 4.35 mg/ml). The organoids were plated in droplets of 40 μl each in a 24-wells plate. After allowing the Matrigel to solidify for 1 h at 37°C, 500 μl pre-warmed complete culture medium (CCM) was added to the organoids. Organoids were passaged once or twice a week.

## Drugs and reagents

APR-246 (2-hydroxymethyl-2-methoxymethyl-1-azabicyclo[2,2,2]octan-3-one), [14]C-labeled APR-246 and glutathione-conjugated MQ (GS-MQ) were obtained from Aprea Therapeutics AB (Stockholm, Sweden). APR-246 was dissolved in either DMSO or water. MK-571 (final concentration to cells 5–20 μM) and Sulfasalazine (final concentration to cells 100–200 μM) were from Merck (Germany). MK-571 for treatment of esophageal cancer cells *in vitro* and *in vivo* was from Selleckchem (Houston, TX) or for colo-PDOs from Cayman Chemical (USA) in 0.9% saline. Reversan (final concentration to cells 10–20 μM) was from Santa Cruz Biotechnology, sc-296262 (USA). Inhibitors were dissolved in DMSO. The percentage of DMSO during drug exposure never exceeded 0.2% for cell lines and 0.5% for eso-PDOs. Doxycycline and chemicals described hereafter were all from Merck (Germany).

## *In vivo* treatments of tumor xenografts

All animal experiments were approved by the Peter MacCallum Cancer Centre (PMCC) Animal Experimentation Ethics Committee and undertaken in accordance with the National Health and Medical Research Council Australian Code of Practice for the Care and Use of Animals for Scientific Purposes. Subcutaneous xenografts from Eso26 cells were established in 6-week-old, female Balb/c nude mice, as previously described (Liu *et al*, 2015). Tumors were measured unblinded using digital calipers every 2–4 days and tumor volume calculated using the formula (length × width$^2$)/2. Mice were randomized to treatment cohorts once tumors reached > 150 mm$^3$ to receive vehicle (0.9% saline + DMSO, $n = 8$), APR-246 (50 mg/kg in 0.9% saline, $n = 8$), MK-571 (50 mg/kg in DMSO, $n = 8$), or APR-246 + MK571 ($n = 9$) administered via intraperitoneal (i.p.) injection. APR-246 or saline (vehicle) was given daily for 16 days, while MK-571 or DMSO (vehicle) was administered daily except on days 6–7 and 13–14. Mice were killed when tumors reached > 1,400 mm$^3$ or at other ethical endpoints (e.g., ulcerating tumor). Time to reach a tumor volume of > 1,400 mm$^3$ was used as a surrogate end-point of survival. Survival of different treatment groups was compared using Kaplan–Meier curves. Mice that were culled at ethical endpoints prior to reaching a tumor volume of > 1,400 mm$^3$ are marked as censored observations.

## Immunohistochemistry

Tumors for immunohistochemistry (IHC) were from mice treated for five consecutive days or 22 days after treatment initiation except for one mouse in the MK-571 only group (24 days) and two mice in the combination group (24 and 32 days). Tumor tissues were formalin-fixed and paraffin-embedded using standard procedures. Tissue sections were deparaffinized and rehydrated, and antigen retrieval was achieved by pressure cooking in citrate buffer (pH 6) for 5 min. Sections were blocked in 10% (*w/v*) bovine serum albumin (BSA) in TBS-Tween-20 0.1% (TBS-T) for 1 h at room temperature and incubated with primary antibodies diluted in 1% (*w/v*) BSA in TBS-T overnight at 4°C. Sections were then incubated with HRP-conjugated secondary antibody (K4003, Dako EnVision + System-HRP labeled Polymer, Dako) for 45 min at room temperature and stained by DAB (K3468, Dako) for 2 min at room temperature. Sections were counterstained with hematoxylin. Images were captured on a BX51 microscope (Olympus) at 20 magnification and visualized using OlyVia V3.2 (Olympus Life Science) software. Primary antibodies used were against Ki67 (1:100, ab16667, Abcam), cleaved caspase 3 (1:100, 9664, Cell Signaling Technology), MRP1 (D5C1X) (1:200, 72202, Cell Signaling Technology), and p53 (1:100, NCL-L-p53-DO7, Leica Biosystems). Quantification of Ki67 and cleaved caspase 3 was assessed by HALO image analysis (Indica Labs) using whole slide multiplex IHC v2.3.4 algorithm.

## Western blotting

Samples were washed in PBS and spun down (5 min at 300 × *g*). Cells were then lysed in ice-cold RIPA buffer (150 mM NaCl, 1% NP40, 0.5% sodium deoxycholate, 0.1% SDS, 50 mM Tris–Cl pH 8), and the insoluble fraction was removed by centrifugation (20,000 × *g* for 15 min at 4°C). Protein concentrations were determined using Bradford protein assay and DC™ Protein assay (Bio-Rad, USA), and absorbance was measured using a cuvette absorbance reader (Bio-Rad, USA) or Tecan microplate reader (Tecan Trading AG, Switzerland). Proteins were heated in LDS samples buffer containing reducing agent and loaded onto NuPAGE 10 % Bis-Tris gels and run in MES or MOPS buffer (all from Thermo Fisher Scientific, USA). Gels were transferred by iBlot (Thermo Fisher Scientific, USA) or wet transfer using Bio-Rad transfer system onto nitrocellulose membrane (Thermo Fisher Scientific, USA) and transfer buffer (25 mM Tris, 192 mM glycine, 20% methanol). Membranes were visualized using Ponceau S (Bio-Rad, USA) in order to assess loading migration pattern. Membranes were subsequently blocked for 1 h at room temperature in PBS-Tween-20 0.1% (PBS-T) containing 5% milk. The nitrocellulose membranes were cut before probing with primary antibodies. Membranes were cut at approx. the 75 kDa size marker and above the 37 kDa size marker. The upper part of the membrane was incubated with an antibody against MRP1, while the middle section was probed with a p53 antibody. The lower part of the membrane was initially incubated with an antibody against xCT (SLC7A11). After visualizing xCT, the

membranes were washed and blotted for GAPDH or β-actin. Primary antibodies used were HRP-conjugated DO-1 or FL-393 against p53 diluted 1:5,000 (sc-126 HRP or sc-6243, respectively, Santa Cruz Biotechnology, USA,), D7O8N against MRP1 diluted 1:1,000 (14685S, Cell Signaling Technology, USA), monoclonal rat anti-MRP1 antibody diluted 1:250 (ALX-801-007-C125, Enzo Life Sciences), D2M7A against xCT/SLC7A11 diluted 1:1,000 (12691S, Cell Signaling, USA), HRP-conjugated antibody against GAPDH diluted 1:30,000 (sc-47724 HRP, Santa Cruz Biotechnology, USA), and antibody against β-actin diluted 1:2,000 (A1978, Merck, Germany). Primary antibodies were incubated in 5% milk overnight at 4°C except if HRP-conjugated, then 1 h at room temperature. HRP-conjugated secondary antibodies against mouse (61-6520, Invitrogen), rat (NA935V, GE Healthcare, UK), or rabbit (65-6120, Invitrogen) IgG were diluted 1:10,000 and incubated for 1 h at room temperature and then visualized using SuperSignal West Femto Maximum Sensitivity substrate (Thermo Fisher Scientific, USA) in a Fuij Film developer (Fuji Film, Japan). Prestained Protein Standards Precision Plus Protein All Blue or Dual Color (Bio-Rad, USA) were used as molecular weight standard.

### siRNA transfections

HCT116 cells ($2.5 \times 10^5$) were plated in 6-well plates and transfected with 10 nM siRNA 6–8 h later using HiPerfect (Qiagen, Germany) according to the manufacturer's protocol. AllStars Negative Control siRNA (Qiagen, Germany) and siGENOME Non-Targeting siRNA (Dharmacon, USA) were used as negative controls. *ABCC1* (MRP1), *TP53,* and *SLC7A11* (xCT) were silenced using four, two, and three different siRNA sequences, respectively (additional details are included in Appendix Table S6). To evaluate the siRNA knockdown efficiency, MRP1, xCT, and p53 protein levels were assessed by Western blotting 24 h after transfection.

### cDNA transfections

TurboFect™ Transfection Reagent (Thermo Fisher) was prepared with plasmid according to the manufacturer's protocol and added to plates. HCT116 cells ($6 \times 10^3$ or $1 \times 10^5$) were added in a 96- or 6-well plate containing transfection mixture with 0.05 μg or 1 μg plasmid, respectively. MRP1/ABCC1 was expressed by pcDNA3.1 (-)-MRP1$_k$ obtained from Susan Cole (Ito *et al*, 2001). xCT/SLC7A11 was expressed using GeneEZ™ SLC7A11 ORF cloneID OHu13066C in pcDNA3.1(+) purchased from GenScript. Empty vector (EV) pcDNA3.1(−) obtained from Susan Cole was used as a negative control. Transfection efficiency was evaluated 72 h later by Western blotting.

### $^{14}$C measurements

For the 6-h time point to assess $^{14}$C-APR-246 accumulation, $5 \times 10^4$ cells were seeded into 96-well plates and treated immediately with the indicated concentrations of $^{14}$C-APR-246. For the 24-h treatment, $6 \times 10^3$ cells were seeded per 96-well and treated directly with 10 or 25 μM $^{14}$C-APR-246 with or without indicated concentrations of MK-571, Reversan and Sulfasalazine. For siRNA experiments, cells were seeded and transfected 24 h prior to re-seeding and $^{14}$C-APR-246 treatment. After incubation for 6 h or 24 h, plates were centrifuged at $200 \times g$ for 5 min. Medium was thereafter removed; the cells were

washed once in PBS, centrifuged, and subsequently trypsinized at 37°C. After detachment of cells, medium was added to wells and cells were transferred onto a printed glassfiber filter mat (PerkinElmer, USA, 1450-421) using a Cell harvester (TOMTEC, USA). Filters with cells were air dried overnight and mounted using melt-on MeltiLex scintillator sheet (PerkinElmer, USA, 1450-441), inserted into a sample bag (Perkin Elmer, USA, 1450-432) and melted using a micro-sealer (Wallac, PerkinElmer, USA). Beta emission from $^{14}$C decay was measured (2 min/96-well) using a MicroBeta Trilux liquid scintillation and luminescence counter (Wallac, 1450, PerkinElmer, USA). $^{14}$C counts per minute (cpm) were measured in $5 \times 10^4$ seeded cells for the 6 h time point. For the 24-h time point, cpm values were normalized by protein concentrations determined using the DC™ Protein assay (Bio-Rad, USA). Protein assay absorbances were measured using a Tecan microplate reader. Protein concentrations were determined from cell culture plates seeded and treated with unlabeled APR-246 in parallel with the $^{14}$C-labeled plates.

### Growth suppression assays in cell lines

Cells ($3 \times 10^3$ except $5 \times 10^3$ for Eso26, FLO-1 and JH-EsoAd1) were seeded in 96-well plates in 100 μl media and treated with indicated compounds after 24 h; growth suppression was determined with either WST-1 substrate (Roche, Switzerland) or resazurin reagent solution. WST-1 was added after 72-h exposure to compounds according to the manufacturers' protocol and absorbance was measured using a Tecan microplate reader. Resazurin reagent solution (655 μM Resazurin, 78 μM Methylene Blue, 1 mM Potassium hexacyanoferrate (III) and 1 mM Potassium hexacyanoferrate (II) trihydrate in PBS) was prepared (all from Sigma), and supplemented to media in a 1:4 ratio. Fluorescence by resazurin was measured by Cytation 3 Cell Imaging Multi-Mode Reader (BioTek). In siRNA experiments, cells were first seeded and transfected with siRNA in 6-well plates, and then re-seeded at $6 \times 10^3$ cells/well in 96-well plates 24 h later. For plasmid transfection, $6 \times 10^3$ cells/well were seeded into a 96-well plate containing transfection mixture. Cells were treated with APR-246 and/or MK-571 the next day. WST-1 substrate was added after treatment for 48 h in siRNA-transfected cells or 72 h for cDNA-transfected cells. Absorbance/fluorescence values representing metabolic activity were expressed as percentage of viable cells compared with untreated cells, or recalculated and presented as percentage of growth suppression relative to the untreated cells.

### Growth suppression assays in PDOs

Cell viability assessment of eso-PDOs was performed as described (Li *et al*, 2018) with modifications. Single eso-PDO cells ($5 \times 10^3$) were seeded in 96-well plate in complete medium (see Appendix Table S2), and drugs were added 72 h later when organoids were formed. Cell viability of eso-PDOs was assessed with CellTiter-Glo (CTG) (Promega) after 6 days of drug treatment, and values were normalized to DMSO-treated cells. Colo-PDOs were seeded as single cells ($1 \times 10^4$), and drugs were added when organoids were formed after 72 h or 96 h (colo-PDO2 and colo-PDO1, respectively). To dissociate PDOs, ice-cold PBS was added to dissolve the Matrigel, cells were then centrifuged at 350 $g$ for 5 min and resuspended in 40% TrypLE Express (12605028, Thermo Fisher Scientific, Waltham, MA, USA) in PBS for 1 h at 37°C. Once

dissociated, cells were centrifuged at 350 *g* for 5 min, resuspended in BCM (see Appendix Table S2), and counted using Countess™ II, Thermo Fisher Scientific, USA. Dissociated single cells were resuspended in Matrigel (Corning® Matrigel® Growth Factor Reduced phenol red-free, BDAA356231, Bio-Strategy, Melbourne, VIC, Australia) (8.7 mg/ml) and plated in droplets of 10 μl in a flat-bottom 96-well plate. Matrigel solidified for 1 h at 37°C then 100 μl pre-warmed CCM was added to the organoids. Cell viability of colo-PDOs was assessed 72 h after drug treatment by 1-h incubation with 5 μg/ml Propidium Iodide (PI, ab14083, Abcam) and 10 μg/ml Hoechst 33342 (B2261-25MG, Sigma-Aldrich). Bright-field and fluorescent images were taken using a Cytation 5 optical plate reader (BioTek, Winooski, VT, USA) at a 4× magnification and four fields per well, with fluorescence filters set up to detect Texas Red and DAPI. After imaging, cell viability was assessed using CellTiter-Glo 3D (CTG, G9681, Promega, Madison, WI, USA) in the same well according to the manufacturer's protocol and luminescence was read using EnSpire Multimode Plate Reader (PerkinElmer, Waltham, MA, USA). Values were normalized to untreated conditions after removal of background signal (wells containing media only). Organoid segmentation based on bright-field images and quantification of organoid area, number, Hoechst and PI texture, and mean Hoechst and PI intensity were performed using CellProfiler software (Version 3.0.0).

### Live cell analysis by IncuCyte®

OVCAR-3 cells ($3 \times 10^3$) were plated in 96-well plates with 100 μl media 24 h prior to treatments. Cells were treated with indicated concentrations of APR-246 +/− MK-571 or Sulfasalazine. Cell growth curves were recorded by the IncuCyte® S3 Live-Cell Analysis System (Essen BioScience, USA). Four images were taken per well every 3 h during 72 h in duplicate wells. The percentage of cell confluence was assessed using IncuCyte S3 Software and normalized to starting timepoint.

### Sub-G1 assay by PI staining

OVCAR-3 cells ($2 \times 10^5$) were seeded in 6-well plates with 2 ml media. After adhesion overnight, cells were treated with different concentrations of APR-246 and/or MK-571. All cells were harvested after 72 h of drug treatments, washed with cold PBS, and pelleted. The cell pellets were resuspended in 1 ml cold PBS, and 1.3 ml of 99% cold ethanol was added dropwise while vortexing samples. The samples were stored overnight at 4°C for fixation. Thereafter, cells were centrifuged at $3,220 \times g$, resuspended, and stained with 0.05 mg/ml Propidium iodide together with 0.25 mg/ml RNAse A at 37°C for 1 h. Cells were analyzed using NovoCyte (Acea Biosciences, USA) flow cytometer, and $10^4$ single events were gated to determine sub-G1 population using NovoExpress software (Acea Bioscience, USA).

### Total glutathione (GSH and GSSG) determination by enzymatic re-cycling assay

GSH measurements were performed as previously described (Tietze, 1969; Eriksson *et al*, 2009). Briefly, for total glutathione (GSH and GSSG) quantification $5 \times 10^5$ cells were homogenized in ice-cold 10 mM HCl. Proteins were precipitated by adding 5-sulfosalicylic acid to a final concentration of 1%. Samples were centrifuged to remove precipitates and supernatants were collected and stored at −20°C until analysis. Samples were incubated in the presence of 5,5'-dithio-bis-[2-nitrobenzoic acid] (DTNB, 0.73 mM), EDTA (4 mM), dihydronicotinamide-adenine dinucleotide phosphate (NADPH, 0.24 mM), 110 mM $NaH_2PO_4$ buffer (pH 7.4), and glutathione reductase (GR) from baker's yeast (1.2 U/ml). The change in absorbance at 412 nm was followed for 5 min using a Versamax microplate reader (Molecular Devices, Switzerland). Total concentration of GSH + GSSG was calculated based on a GSH standard curve. Samples were not alkylated before measurement. Thus, this protocol does not distinguish between disulfide species (GSSG) and reducing GSH species.

### Analysis of *in vitro* retro Michael addition reactions

Glutathione-conjugated MQ (GS-MQ) (100 μM) in 20 mM ammonium format buffer (pH 7.4) was incubated up to 24 h with equimolar concentrations of NAC at 37°C. At indicated time points, a 50 μl aliquot of the reaction was transferred to −20°C and stored until the last time point of the experiment was collected. Subsequently, samples were analyzed on a Waters Alliance HPLC system operated with MassLynx software (Waters, Sweden), equipped with 2998 Photodiode array detector (Waters, Sweden) and ACQUITY QDa Performance MS detector (Waters, Sweden) employing electrospray ionization technique. A positive ion mode was used for detection and quantification analysis. A LUNA C18 (2), 150 × 3 mm, 3 μm particle size, Penomenex (Værløse, Denmark) column equipped with precolumn and precolumn filter, maintained at 40°C, was used for the analysis. The injection volume was 10 μl. Separation was performed employing a gradient elution with water that was mixed with HPLC grade acetonitrile. Formic acid (HPLC grade) was added to all solvents to a final concentration of 0.1%. The initial elution was isocratic with 1% acetonitrile for 2 min, followed by a linear acetonitrile gradient (1–99%) for 13 min.

### LC-MS analysis of metabolites extracted from cells and cell culture media

OVCAR-3 cells ($2 \times 10^5$) were seeded in 6-well plates with 2 ml media. After allowing the cells to adhere overnight, cells were treated with the indicated concentrations of APR-246 and/or MK-571. After 24-h treatments with compounds, cells were washed and harvested by trypzination. Cell viability was assessed by trypan blue (NanoEntek, USA) staining and cells were counted using an automated cell counter (Countess™ II, Thermo Fisher Scientific, USA). Cells were carefully washed with PBS to remove residues of culture media, lysed using a hypotonic buffer of 20% PBS ($Ca^{2+}/Mg^{2+}$ free), and subjected to freeze-thaw cycles. Proteins were precipitated from cell lysates using ice-cold acetone, at a sample to acetone ratio of 1:4, and incubated overnight at −20°C. Cell debris and proteins were removed by centrifugation. Supernatants, containing low molecular weight molecules, were concentrated by SpeedVac (Savant™, Thermo Fisher Scientific) under vacuum, and concentrated samples were re-constituted to a volume of 30–50 μl with water. Formic acid was added to each sample to a final concentration of 0.1%. Samples were analyzed by LC-MS as described above.

**Measurement of reduced (free) thiol with thiol tracker**

Cells ($0.1 \times 10^6$) were stained in 1 µM Thiol tracker Violet (Life technologies) for 30 min at 37°C. Cells were analyzed using Novo-Cyte flow cytometer, and $10^4$ single events were used for determining geometric mean fluorescence by NovoExpress software.

**Immunofluorescence imaging**

Sterilized coverslips were pre-coated with 0.01% Poly-L-Lysine (P4832, Sigma) for 5 min and then washed with PBS and dried for 2 h. $1 \times 10^5$ HCT116 R248W cells were added and incubated for 72 h. Cells were fixed for 15 min with Pierce™ Methanol-free 16% Formaldehyde (Thermo Fisher) diluted to 4%, washed with PBS, permeabilized 2 min with 0.2% Triton X, washed, and blocked for 60 min. Blocking buffer contained 2% Bovine Serum Albumin (BSA), 5% glycerol, and 0.2% Tween-20 in PBS. Primary antibody D7O8N against MRP1 diluted 1:200 (14685S, Cell Signaling, USA) was prepared in blocking buffer and incubated at 4°C over night. Coverslips were washed with PBS, anti-rabbit (A-110008, Thermo Fisher) and anti-rat (A-110006, Thermo Fisher) Alexa Flour® 488 secondary antibodies were diluted 1:500 in blocking buffer together with Phalloidin-Atto 647N 1:500 (65906-10NMOL, Sigma) and incubated for 1 h. Coverslips were washed and mounted with VECTA-SHIELD HardSet Antifade Mounting Medium with DAPI (H-1500 Vector Laboratories). Next day samples were imaged by Zeiss AxioObserver Z1-inverted microscope equipped with Axiocam 506 mono camera using the 63× oil immersion lens and processed using ZEN software by Zeiss.

**Analysis of the TCGA dataset**

Data from the luad, lusc, coad, and esca studies of The Cancer Genom Atlas (TCGA) PanCancer Atlas Studies (Hoadley *et al*, 2018) were downloaded from cBioportal (Cerami *et al*, 2012; Gao *et al*, 2013) (https://www.cbioportal.org/). Patients were selected for having genetic alternations with putative *TP53* mutation (missense or truncating mutations) or no *TP53* alteration. *ABCC1* mRNA Expression, RSEM (Batch normalized from Illumina HiSeq_RNA-SeqV2), was plotted based on *TP53* alterations in indicated studies. One outlier lung cancer patient was excluded.

**IC$_{50}$ and synergy calculations**

IC$_{50}$ values in cell lines were calculated using Origin software (OriginLab) based on the mean WST-1 data (highest treatment was below 30 µM APR-246) and resazurin assay for esophageal cell lines (up to 100 µM APR-246). In cases for the WST-1 dataset where IC$_{50}$ values were not reached within the concentration range used, data were extrapolated using Origin with a maximum concentration at 30 µM. IC$_{50}$ values of PDOs were estimated from dose response curves generated by an application available at https://vladjnbykov.shinyapps.io/IC50/ which deploys log-logistic regression models from R DRC package (DRM). Synergistic effects by combination treatments in cell lines were determined using the web application tool SynergyFinder 1.0 or 2.0 (https://synergyfinder.fimm.fi/) (Ianevski *et al*, 2017; Ianevski *et al*, 2020). Synergy in PDOs was assessed based on CTG data as well as imaging readouts such as organoid

**The paper explained**

**Problem**

Tumor suppressor *TP53* is the most commonly mutated gene in cancer. *TP53* mutation is associated with poor prognosis and chemoresistance. APR-246 (Eprenetapopt) is the only compound in clinical development that reactivates mutant p53 protein, at Phase III clinical stage in myelodysplastic syndrome (MDS). Further studies of the mechanism of action of APR-246 and potential synergies with other compounds are of great interest in order to increase therapeutic efficacy and expand the therapeutic window. This, along with a better understanding of APR-246 pharmacodynamics and prediction of tumor sensitivity, may facilitate a broader clinical implementation of APR-246.

**Results**

MQ, the active conversion product of APR-246, reversibly binds the major antioxidant glutathione (GSH). We have shown that this complex can be exported through the multidrug resistance-associated protein 1 (MRP1) efflux pump and that blocking MRP1 traps MQ inside the cell. This leads to intracellular accumulation of MQ bound to glutathione, forming a reservoir of MQ that can bind mutant p53 and other cellular targets. Blocking MRP1 also disrupts antioxidant shields often elevated in cancer cells, thus enhancing cancer cell death *in vitro*, *ex vivo*, and *in vivo*.

**Impact**

Our data explain the dynamics of APR-246 and MQ in cancer cells and indicate that MRP1 can play a key role in the sensitivity to APR-246. Altogether our findings suggest that combination treatment with APR-246 and drugs that target the antioxidant balance may allow more efficient cancer therapy.

area and Hoechst and PI intensity. We wrote a customized R script based on the "Synergyfinder" package (https://doi.org/10.18129/B9.bioc.synergyfinder). Both the SynergyFinder R package and the web application tool apply four well-known reference models: Zero interaction potency (ZIP) model (Yadav *et al*, 2015), the highest single agent (HSA) model (Berenbaum, 1989), Bliss independence model (Bliss, 1939) and Loewe additivity model (Loewe, 1953; Lederer *et al*, 2018). Synergy landscape plots over dose matrix were generated by SynergyFinder. Positive, zero, and negative synergy scores indicate synergy, additivity, and antagonism, respectively.

**Data visualization and statistical calculations**

Results are presented as mean ± standard error of the mean (SEM). GraphPad Prism 8 was used to prepare histograms, heatmaps, scatter plots, statistical tests, and correlation and linear regression analysis. Normal distribution was tested by Shapiro–Wilk test. When distribution was normal, paired or unpaired *t*-test was applied. If data did not pass the Shapiro–Wilk test, a non-parametric test was applied; for unpaired data, the Mann–Whitney test was used; for paired data, the Wilcoxon test was applied. Tumor growth volumes at day 18 were compared using one-way ANOVA with Tukey's correction. Survival curves were compared using Log-rank (Mantel–Cox) test of individual treatment groups. PermutMatrix was used to generate heatmap and hierarchical clustering in Appendix Fig S1B. For correlation analysis, the compound activity profile of PRIMA-1, gene expression, and metabolite abundance data was obtained from

Cancer Dependency Map (https://www.depmap.org) to determine potential associations. Pearson correlations between PRIMA-1 area-under-the-curve (AUC) and gene expression or metabolite abundance was calculated using cor.test function in R (ver. 3.6.0) for 708 cancer cell lines. Adobe Illustrator CS6 or 2020 was used to put together figures. If required for better visualization, Adobe Photoshop 2020 was used for image processing. Any changes made in images were applied equally over the entire image and all images compared.

## Data availability

This study includes no data deposited in external repositories.

**Expanded View** for this article is available online.

### Acknowledgements

This work was supported by research grants to K.G.W. from the Swedish Research Council (Vetenskapsrådet), the Swedish Cancer Society (Cancerfonden), the Swedish Childhood Cancer Fund (Barncancerfonden), Radiumhemmets Forskningsfonder, Knut and Alice Wallenberg Foundation, Aprea Therapeutics and Karolinska Institutet, and by a National Health and Medical Research Council (NMHRC) Project Grant #APP1120293 and a Fellowship (MCRF16002) from the Department of Health and Human Services acting through the Victorian Cancer Agency, Victoria, Australia, to N.J.C.. F.H. is supported by a Senior Research Grant from the Tour de Cure Foundation and a Project Grant from the National Health and Medical Research Council of Australia (GNT1164081). We thank Rebecca Fitzgerald (R.C.F.) for her generous funding of the eso-PDO experiments. R.C.F. is funded by a Core Programme Grant from the Medical Research Council (RG84369). Sample collection for organoid generation was funded by a program grant to R.C.F. from Cancer Research UK (RG81771/84119). The Victorian Centre for Functional Genomics (K.J.S.) is funded by the Australian Cancer Research Foundation (ACRF), Phenomics Australia (PA) through funding from the Australian Government's National Collaborative Research Infrastructure Strategy (NCRIS) program, the Peter MacCallum Cancer Centre Foundation and the University of Melbourne Research Collaborative Infrastructure Program (MCRIP). We thank Susan Cole for sharing the MRP1k-pcDNA3.1(-) and relevant plasmids, and Styrbjörn Byström (Aprea Therapeutics) for valuable advice on the reversible nature of MQ conjugation. We are grateful for the technical advice concerning $^{14}$C measurements from Kristina Witt (Karolinska Institutet). We thank Peter Chumakov (Engelhardt Institutet of Molecular Biology) for H1299 cells, Bert Vogelstein (Johns Hopkins Oncology Center) for HCT116 cells, and Rolf Kiessling (Karolinska Institutet) for KADA cells. Finally, we thank Lars-Gunnar Larsson (Karolinska Institutet) for whole exome sequencing and former laboratory member Fredrik Jerhammar for culturing ESTDAB-140, ESTDAB-037, ESTDAB-049, SKMEL-2, and A375 cells. Where indicated, analyses were based on data generated from the TCGA Research Network: https://www.cancer.gov/tcga.

### Author contributions

Conception and design: SC, SEE, LA, KGW; Development of methodology: SC, SEE, VJNB, MG, XL, CB, NJC; Acquisition of data: SC, SEE, EHC, SD, MCB, VJNB, KMF, MG, XL, NJC; Analysis and interpretation of data: SC, SEE, EHC, SD, VJNB, KMF, MG, XL, SR, KJS, FH, LA, NJC, KGW; Writing of the manuscript: SC, SEE, KGW; Review of the manuscript: SC, SEE, EHC, SD, MC, VJNB, KMF, MG, XL, SR, CB, KJS, FH, LA, NJC, KGW; Study supervision: SC, SEE, NJC, KGW; Funding acquisition: KJS, FH, NJC, KGW.

### Conflict of interest

This study was funded in part by Aprea Therapeutics, a company that develops p53-based cancer therapy, including APR-246. K.G.W. and V.J.N.B. are co-founders and shareholders of Aprea Therapeutics. K.G.W. is also a member of its Clinical Advisory Board. K.G.W. and V.J.N.B. have received a salary from Aprea Therapeutics. L.A. is Chief Scientific Officer of Aprea Therapeutics.

### For more information

Public data were downloaded from https://www.cbioportal.org/ and https://www.depmap.org.

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
