## [Review Process File · EMBO Molecular Medicine]

A thiol-bound drug reservoir enhances APR-246-induced mutant p53 tumor cell death

Sophia Ceder, Sofi Eriksson, Emarndeena Cheteh, Swati Dawar, Mariana Corrales Benitez, Vladimir Bykov, Kenji Fujihara, Mélodie Grandin, Xiaodun Li, Susanne Ramm, Corina Behrenbruch, Kaylene Simpson, Frederic Hollande, Lars Abrahmsen, Nicholas Clemons, and Klas Wiman

DOI: [10.15252/emmm.201910852](https://doi.org/10.15252/emmm.201910852)

Corresponding authors: Klas Wiman (Klas.Wiman@ki.se)

Review Timeline:

Submission Date:	7th May 19
Editorial Decision:	8th May 19
Revision Received:	22nd Jan 20
Editorial Decision:	13th Feb 20
Revision Received:	15th Oct 20
Editorial Decision:	3rd Nov 20
Revision Received:	16th Nov 20
Accepted:	17th Nov 20

Editor: Celine Carret / Lise Roth

Transaction Report:

8th May 2019

Decision on your manuscript EMM-2019-10852

Dear Dr Wiman,

Thank you for submitting your manuscript "Disrupting redox homeostasis in cancer cells enhances efficacy of mutant p53 reactivator APR-246" to EMBO Molecular Medicine. I have now carefully read your manuscript and discussed it with my colleagues. I regret to say that we find the manuscript not well suited for publication in EMBO Molecular Medicine and therefore have decided not to proceed with peer review.

The manuscript convincingly describes how MRP1 inhibition synergises with APR-246 to induce cancer cell death in several tumour cell lines. Mechanistically, MRP1 inhibition shifts GSH and cysteine concentrations within cells and induces the accumulation of GS-MQ, which acts as a reservoir for further target interactions. While the study is of interest and suggests a combinatorial therapy to improve the efficacy of APR-246, we deeply regret that no in vivo data are shown to confirm the pathophysiological relevance and anti-cancer effect in an appropriate model. Without this, I am afraid that we cannot offer to consider the article further as our journal focuses on clinical significance and translational implications. However, should you be able to add in vivo data, I would be happy to reconsider this decision.

But for now, I am sorry to have to disappoint you; in the interest of time, I am providing you with an early decision that will allow you to submit your manuscript elsewhere without any further delays.

Please rest assured that this is not a judgment of the quality or interest of your work but a decision based on appropriateness for EMBO Molecular Medicine.

Yours sincerely,

Celine Carret

Celine Carret, PhD
Senior Editor
EMBO Molecular Medicine

As a service to authors, EMBO provides authors with the possibility to transfer a manuscript that one journal cannot offer to publish to another EMBO publication. The full manuscript and if applicable, reviewers reports are automatically sent to the receiving journal to allow for fast handling and a prompt decision on your manuscript. For more details of this service, and to transfer your manuscript to another EMBO title please click on Link Not Available

To the Editorial Office, EMBO Molecular Medicine

Dear Dr. Celine Carret,

On behalf of all authors, I would like to kindly ask you to re-consider the enclosed manuscript entitled "A thiol-bound drug reservoir enhances APR-246-induced mutant p53 tumor cell death" for publication in EMBO Molecular Medicine. Based on your request we have performed mouse experiments with APR-246 and MRP1 inhibitor MK-571.

This manuscript presents novel findings with fundamental importance for APR-246 drug pharmacodynamics and induction of tumor cell death. APR-246 is currently tested in a phase III clinical study in a poor prognosis subgroup of myelodysplastic syndrome (MDS) patients with mutant *TP53*, and in several phase II clinical trials. The *TP53* gene is mutated in approximately 50% of all tumors and *TP53* mutation is associated with chemotherapy resistance and lower overall survival rates. MDS patients carrying mutant *TP53* have particularly poor prognosis and are in desperate need of new effective therapy options.

We have previously discovered and characterized the mutant p53-targeting compound APR-246 and its active product MQ, which is a Michael acceptor (Bykov et al., Nature Med. 8, 282-8, 2002; Lambert et al., Cancer Cell 15, 376-388, 2009). APR-246 is the most advanced mutant p53-targeting compound and represents a whole new drug class. Therefore, major efforts in the field are now being devoted to understanding the full molecular mechanism of action of APR-246. We have previously shown that MQ not only targets mutant p53 but also cellular antioxidant systems, an Achilles's heel in many cancers.

Our attached manuscript presents three novel important findings that explain how APR-246 triggers tumor cell death: 1) We show that intracellular thiol status in general, including levels of both glutathione and cysteine, is critical for the efficacy of APR-246; 2) We demonstrate that the covalent conjugation of MQ to a cellular thiol such as glutathione is rapidly reversible; and 3) We show that the reversible MQ binding to thiols creates a reservoir of active drug within the tumor cell. This MQ reservoir is regulated by the cellular thiol content as well as MRP1 efflux pumps. Hence, blocking the cellular efflux of GS-MQ and/or depleting thiol-based antioxidants can greatly potentiate the efficacy of APR-246-induced tumor cell death *in vivo* and *in vitro* by increasing the amount of intracellular MQ available for binding mutant p53 and other protein targets. Thus, the intrinsic reversibility of MQ reactivity that we have discovered is a key aspect of the mode of action of APR-246. This reversibility most likely contributes to the benign safety profile of APR-246 as shown by clinical studies.

These findings may also have general relevance for understanding efficacy of other Michael acceptors already approved for clinical use, for instance ibrutinib that targets Bruton's kinase

and osimertinib that targets the epidermal growth factor receptor (reviewed by Jackson et al., J. Med. Chem. 60, 839-85, 2017).

Moreover, our results may have direct clinical relevance, since they suggest that combination treatment with APR-246 and agents that perturb antioxidant balance could greatly enhance anti-tumor efficacy of APR-246. One example is the FDA-approved drug sulfasalazine. Thus, our study not only presents novel information about mode of action of APR-246 at the molecular and cellular level, but may also have important implications for the future use of this drug in the clinic.

All authors have read and approved submission of this manuscript. We look forward to hearing from you soon.

13th Feb 2020

Dear Dr. Wiman,

Thank you for the submission of your manuscript to EMBO Molecular Medicine. We have now heard back from the three referees whom we asked to evaluate your manuscript.

You will see that all referees find the study to be of interest. While referees #1 and #2 are rather supportive, referee #3 is more critical, in particular regarding the *in vivo* model provided that is not appropriate. This referee confirms during our cross-commenting exercise that should you provide a convincing PDX model with clinical relevance, the manuscript would be greatly strengthened and a good candidate for publication.

We would therefore welcome the submission of a revised version within three months for further consideration and would like to encourage you to address all the criticisms raised as suggested to improve conclusiveness and clarity. Please note that EMBO Molecular Medicine strongly supports a single round of revision and that, as acceptance or rejection of the manuscript will depend on another round of review, your responses should be as complete as possible.

I look forward to receiving your revised manuscript.

Yours sincerely,

Celine Carret

Celine Carret, PhD
Senior Editor
EMBO Molecular Medicine

*** Instructions to submit your revised manuscript ***

**** PLEASE NOTE **** As part of the EMBO Publications transparent editorial process initiative (see our Editorial at <https://www.embopress.org/doi/pdf/10.1002/emmm.201000094>), EMBO Molecular Medicine will publish online a Review Process File to accompany accepted manuscripts.

To submit your manuscript, please follow this link:

Link Not Available

- 1) a .doc formatted version of the manuscript text (including Figure legends and tables). Please make sure that the changes are highlighted to be clearly visible to referees and editors alike.
- 2) separate figure files*
- 3) supplemental information as Expanded View and/or Appendix. Please carefully check the authors guidelines for formatting Expanded view and Appendix figures and tables at <https://www.embopress.org/page/journal/17574684/authorguide#expandedview>
- 4) a letter INCLUDING the reviewers' reports and your detailed responses to their comments (as Word file)

Also, and to save some time should your paper be accepted, please read below for additional information regarding some features of our research articles:

- 5) The paper explained: EMBO Molecular Medicine articles are accompanied by a summary of the articles to emphasize the major findings in the paper and their medical implications for the non-specialist reader. Please provide a draft summary of your article highlighting
 - the medical issue you are addressing,
 - the results obtained and
 - their clinical impact.

- 6) For more information: There is space at the end of each article to list relevant web links for further consultation by our readers. Could you identify some relevant ones and provide such information as well? Some examples are patient associations, relevant databases, OMIM/proteins/genes links, author's websites, etc...

7) Author contributions: the contribution of every author must be detailed in a separate section (before the acknowledgments).

8) EMBO Molecular Medicine now requires a complete author checklist (<https://www.embopress.org/page/journal/17574684/authorguide>) to be submitted with all revised manuscripts. Please use the checklist as a guideline for the sort of information we need WITHIN the manuscript as well as in the checklist. This is particularly important for animal reporting, antibody dilutions (missing) and exact p-values and n that should be indicated instead of a range.

9) Every published paper now includes a 'Synopsis' to further enhance discoverability. Synopses are displayed on the journal webpage and are freely accessible to all readers. They include a short stand first (maximum of 300 characters, including space) as well as 2-5 one sentence bullet points that summarise the paper. Please write the bullet points to summarise the key NEW findings. They should be designed to be complementary to the abstract - i.e. not repeat the same text. We encourage inclusion of key acronyms and quantitative information (maximum of 30 words / bullet point). Please use the passive voice. Please attach these in a separate file or send them by email, we will incorporate them accordingly.

You are also welcome to suggest a striking image or visual abstract to illustrate your article. If you do please provide a jpeg file 550 px-wide x 400-px high.

10) A Conflict of Interest statement should be provided in the main text

11) Please note that we now mandate that all corresponding authors list an ORCID digital identifier. This takes <90 seconds to complete. We encourage all authors to supply an ORCID identifier, which will be linked to their name for unambiguous name identification.

Currently, our records indicate that there is no ORCID associated with your account.

Please click the link below to provide an ORCID:

Link Not Available

12) The system will prompt you to fill in your funding and payment information. This will allow Wiley to send you a quote for the article processing charge (APC) in case of acceptance. This quote takes into account any reduction or fee waivers that you may be eligible for. Authors do not need to pay any fees before their manuscript is accepted and transferred to our publisher.

Photos 400-800 DPI

Figures are not edited by the production team. All lettering should be the same size and style; figure panels should be indicated by capital letters (A, B, C etc). Gridlines are not allowed except for log plots. Figures should be numbered in the order of their appearance in the text with Arabic numerals.

Each Figure must have a separate legend and a caption is needed for each panel.

*Additional important information regarding figures and illustrations can be found at <http://bit.ly/EMBOPressFigurePreparationGuideline>

***** Reviewer's comments *****

Referee #1 (Remarks for Author):

The MS entitled "A thiol-bound drug reservoir enhances APR-246-induced mutant p53 tumor cell death." by Ceder, Eriksson et al., focuses on the mechanism of action of APR-246 in cancer cells in the context of those processes that control drug efflux and intracellular thiol pools.

By performing biochemical, cellular, bioinformatics and in vivo experiments the authors have clearly established that APR-246 sensitivity of cancer cells is associated to the capability to retain and accumulate the APR-246 byproduct and Michael acceptor MQ and to the cellular pool of the thiols GSH and Cysteine. Thus, a synergistic growth suppressive activity of APR-246 is obtained by using inhibitors for MRP1 and xCT that block the export of MQ-thiol conjugates and the import of Cysteine groups, respectively.

Even though missense mutants of p53 are the primary target of APR-246, and the synergy between APR-246 and MRP1 inhibitor MK-571 was more prominent in cells carrying mutant TP53 and the authors showed that no obvious correlation exists between APR/MQ accumulation and TP53 status. Thus the authors are aware that additional factors besides total MQ accumulation and TP53 status determine the extent of synergy.

The MS is linear and of interest considering the impact of out p53 in cancer. In general, the experiments are well performed and sustain the conclusions drawn. However there are some concerns that could help clarify unclear issues.

Concerns:

- (Figs. S1D and 6D) It is not clear from the Western blot analysis which band corresponds to the 190 KDa MRP1. Thus, in the opinion of this reviewer, a better understanding of this issue might also lead to a different conclusion about the association with APR-246 IC50. These analyses should be done with another antibody and should be accompanied by immunofluorescence analyses.
- Furthermore in Fig. 6D last lane, it is not clear why there is not a dose-dependent increase of xCT levels after increasing amounts of APR-246 in MK-571 (μ M) treated cells.
- An added value would also be the imaging, at the cellular level, of the accumulation of MQ and the change in LMW thiol groups upon inhibition of the two transporters.
- Moreover, does MRP1 up-regulation after knockdown rescue resistance to APR-246?
- Moreover (Fig. 1C) Among the "APR-246 high sensitivity" cell lines there is a p53 WT cell line A375 with a very low synergy index. Could you explain or make a hypothesis?

Referee #2 (Comments on Novelty/Model System for Author):

This work describes the effects of combined treatment with APR-246, a mutant p53 targeting drug, with inhibition of the efflux pump MRP1/ABCC1. The potential mechanism of action of APR-246 has been described extensively, involving MQ binding to cysteine residues. It has also been described

that APR-246 (MQ) affects GSH and TXN antioxidant systems including the role of SLC7A11. Finally, an important role for MRP1/ABCC1 in redox regulation by controlling GSH/GSSG homeostasis has been described before. The work described here integrates this knowledge in the combined and enhanced effects of inhibition of the different processes. From this work it is difficult to pinpoint the observed effects to a single mechanism as they strongly affect each other and all seem to contribute to the enhanced killing.

The model systems used are relevant and extent to in vivo analysis, the technical quality of the experiments is high and supports the conclusions in this manuscript.

With respect to the medical impact, the correlation with mutant p53 is not very strong and it seems that the drug combination is not specific for mutant p53 cancers. This could potentially limit the clinical use and impact.

Referee #2 (Remarks for Author):

The manuscript is well written, the experiments well performed and they strongly support the conclusions drawn from this work. As the authors state on page 4, lines 9-13, it remains difficult to separate the different parameters and therefore reach a strong conclusion on the contribution of the different cellular changes to the enhanced killing. However, the work present here is interesting and highlights the importance of redox homeostasis for the effects of APR-246.

Minor points for improvement:

1. Figure 3B, it would be informative to reflect the changes for each specific cell cline rather than a gift in overall behavior.
2. In figure S4C, it seems that the equilibrium is more towards NAC-MQ while the reaction arrows suggest an equal distribution?
3. The conclusions on page 7 line 17 and page 8 lines 11-12 seem to contradict each other. This should be corrected or more clearly explained.
4. The statement on page 8, lines 19-20 does not seem to be supported by Figure 1D-E, where mutant and null seem not significantly different.

Referee #3 (Comments on Novelty/Model System for Author):

The main problem of the manuscript is the lack of robust in vivo experiments which are essential to validate the clinical value of the reported findings. The use of PDX of diverse types (at least two) has been indicated to the authors. Since APR-246 is already in clinical trials any additional co-treatment which should potentially reduced its toxicity and increase its efficacy needs to be tested as first step in in vivo experiments and only as a second step in cell lines. This main flaws makes the manuscript unsuitable for publication in EMBO MOL MED.

Referee #3 (Remarks for Author):

Wiman's group reports that pharmacological inhibition or siRNA-mediated depletion of MRP1 increases the anticancer activity of APR-246, a reactivator of mutant p53 proteins in human cancers. APR-246 clinical efficacy is actually tested in different clinical trials and the reported findings might represent the rationale for new therapeutic approaches of human cancer carrying

TP53 mutations and expressing gain of function mutant p53 proteins.

While the pre-clinical and biochemical data are well designed and performed the lack of a broad range of in vivo experiments (for instance PDX derived from tumors carrying either TP53 mutations or intact TP53 gene) strongly flaws the overall message of the manuscript.

Indeed, the reported data provide additional evidence of the potential use of APR-246 in combination treatment but they do not provide any robust in vivo evidence that can support translation to clinical application.

Specific comments

1. Is there any correlation between TP53 status and expression of MRP1 in TGCA of different human cancers?
2. Pag.5 lines 22: "MRP1 protein expression did not show any obvious association to TP53 gene status, APR-246, IC50, or d-score (Fig. S1D)". This sentence refers to cell lines, but it seems to be in contrast with the rationale of the work that MRP1 could have a role in redox regulation and export of GSH-conjugated drugs increasing the resistance to PRIMA-1 only in a mutp53 cellular background (Fig. 1E).
3. Protein expression analysis upon MK-571 treatment needs to be performed in different cell lines and not only for OVCAR-3.
4. The only set of in vivo experiments was performed by using Eso26 cells (no IC50 was identified). What about synergy with APR-246? This cell lines was not characterized along the manuscript. These experiments need to be performed by using different cell lines as some of those analyzed in Fig. 1.
5. IHC for p53, MRP1 and ki67 expression needs to be performed in the xenografted tumors.
6. Figure 6 shows the shifts cellular thiol pools by MRP1 inhibitor MK-571 seem to be rather modest.

Dear Dr. Wiman,

Thank you for the submission of your manuscript to EMBO Molecular Medicine. We have now heard back from the three referees whom we asked to evaluate your manuscript.

You will see that all referees find the study to be of interest. While referees #1 and #2 are rather supportive, referee #3 is more critical, in particular regarding the in vivo model provided that is not appropriate. This referee confirms during our cross-commenting exercise that should you provide a convincing PDX model with clinical relevance, the manuscript would be greatly strengthened and a good candidate for publication.

We would therefore welcome the submission of a revised version within three months for further consideration and would like to encourage you to address all the criticisms raised as suggested to improve conclusiveness and clarity. Please note that EMBO Molecular Medicine strongly supports a single round of revision and that, as acceptance or rejection of the manuscript will depend on another round of review, your responses should be as complete as possible.

I look forward to receiving your revised manuscript.

Yours sincerely,

Celine Carret

Celine Carret, PhD
Senior Editor
EMBO Molecular Medicine

Point-by-point response to referee comments

***** Reviewer's comments *****

Referee #1 (Remarks for Author):

The MS entitled "A thiol-bound drug reservoir enhances APR-246-induced mutant p53 tumor cell death." by Ceder, Eriksson et al., focuses on the mechanism of action of APR-246 in cancer cells in

the context of those processes that control drug efflux and intracellular thiol pools.

By performing biochemical, cellular, bioinformatics and in vivo experiments the authors have clearly established that APR-246 sensitivity of cancer cells is associated to the capability to retain and accumulate the APR-246 byproduct and Michael acceptor MQ and to the cellular pool of the thiols GSH and Cysteine. Thus, a synergistic growth suppressive activity of APR-246 is obtained by using inhibitors for MRP1 and xCT that block the export of MQ-thiol conjugates and the import of Cysteine groups, respectively.

Even though missense mutants of p53 are the primary target of APR-246, and the synergy between APR-246 and MRP1 inhibitor MK-571 was more prominent in cells carrying mutant TP53 and the authors showed that no obvious correlation exists between APR/MQ accumulation and TP53 status. Thus the authors are aware that additional factors besides total MQ accumulation and TP53 status determine the extent of synergy.

The MS is linear and of interest considering the impact of out p53 in cancer. In general, the experiments are well performed and sustain the conclusions drawn. However there are some concerns that could help clarify unclear issues.

Concerns:

(Figs. S1D and 6D) It is not clear from the Western blot analysis which band corresponds to the 190 KDa MRP1. Thus, in the opinion of this reviewer, a better understanding of this issue might also lead to a different conclusion about the association with APR-246 IC50. These analyses should be done with another antibody and should be accompanied by immunofluorescence analyses.

Author response We agree with the referee that the MRP1 migration pattern on the Western blots is somewhat confusing. MRP1 protein is N-glycosylated with complex-type oligosaccharides on asparagine sites (Hipfner, Almquist et al., 1997). Different cell lines may have different glycosylation patterns, resulting in smears and multiple MRP1 bands (Hipfner et al., 1997, Pillai-Kastoori, Heaton et al., 2020, Shukalek, Swanlund et al., 2016, Stride, Grant et al., 1997). Mutations of MRP1 glycosylation sites or treatment with a de-glycosylation enzyme can change the migration of MRP1 bands on the blot (Hipfner et al., 1997, Shukalek et al., 2016, Stride et al., 1997). Therefore, we do not expect that MRP1 will appear as a single band and also that the sizes of the MRP1 bands will vary between cell lines, as shown in Fig. S1D. According to Cell Signalling (c.s), the manufacturer of the antibody used for the first submission of the manuscript, the epitope recognizes MRP1 migrating in the range of 170-220kDa.

To clarify which Western blot bands correspond to MRP1 we have repeated Western blot analysis of HCT116 TP53 WT and R248W cells transfected with four siRNA sequences targeting MRP1. Protein lysates were analyzed using two different antibodies for MRP1, i.e. the c.s. antibody plus a previously published antibody (Gana, Hanssen et al., 2019) from Enzo Life Science (enzo). We have replaced the previous Figure S3E with the Western blots now included in in Fig. 1H and Appendix Fig. S1I (below).

We have also transfected cells with the pcDNA3.1(-)/MRP1_k plasmid (MRP1) and its empty vector (EV) to further validate the two antibodies (c.s. and enzo), as shown in the extra figure below. Although the antibodies give rise to slightly different band patterns, both antibodies detect one or several protein bands within the expected size region of MRP1. Furthermore, all bands were decreased upon siRNA-mediated MRP1 knockdown and increased upon transfection with MRP1 plasmid.

Figure 1**Appendix Figure S1****Fig. 1H** Western blot analysis of MRP1 (c.s.), p53 (DO-1) and GAPDH of HCT116 WT and R248W cells 48h after transfection with negative control siRNAs or siRNAs targeting MRP1 or p53.**Fig. S1I** Western blot analysis with another MRP1 antibody (enzo), p53 (DO-1) and GAPDH of HCT116 WT and R248W cells 48h after transfection with negative control siRNAs or siRNAs targeting MRP1 or p53.**Extra figure**
As requested by the referee, we have also added immunofluorescence (IF) images using the IF-validated c.s. antibody in Appendix Fig. S1E to confirm that the detected MRP1 is indeed localized at the cell membrane as shown by co-staining with Phalloidin which binds filamentous actin. Additional images of samples without Phalloidin staining which are not included in the manuscript are shown below.

Appendix Figure S1

Fig. 1E Membrane localization of MRP1 (c.s.) and Phalloidin immunofluorescence staining of HCT 116 R248W cells.

Extra figure

Membrane localization of MRP1 (c.s.) immunofluorescence staining of HCT 116 R248W cells.

Furthermore in Fig. 6D last lane, it is not clear why there is not a dose-dependent increase of xCT levels after increasing amounts of APR-246 in MK-571 (μM) treated cells.

Author response: Figure 6D has been updated with a Western blot which shows a dose-dependent effect of 0-10 μM APR-246 in combination with MK-571. The apparent decrease at 20 μM APR-246 and MK-571 may be due increased toxicity at this combination even though cells still exclude trypan blue (Fig. S4A). At this dose cells may have entered a pre-apoptotic stage which could explain the decrease in GAPDH. Furthermore, we have included Western blot analysis of two more cell lines as requested by Referee #3.

Figure 6

D

Fig. 6D Western blot of OVCAR-3 cells treated with APR-246 +/- MK-571 for 24h.

An added value would also be the imaging, at the cellular level, of the accumulation of MQ and the change in LMW thiol groups upon inhibition of the two transporters.

Author response: MQ is a Michael acceptor with a double bond that reacts covalently with cysteines. As demonstrated here and in our previous studies (Haffo, Lu et al., 2018, Lambert, Gorzov et al., 2009) these bonds are reversible and dynamic, and affected by shifts in cellular redox homeostasis. Although we agree with the referee that it would be an added value to image MQ accumulation and changes in thiol groups in cells, this is technically challenging due to high reactivity of MQ and the reversible nature of the covalent bond. MQ is a very small molecule therefore modification with for example a fluorophore to allow imaging is technically challenging and might affect the reactivity of the molecule. The use of another thiol-reactive probe for imaging would influence the intracellular dynamics of MQ covalent bond formation.

However, changes in LMW thiol groups after inhibition of MRP1 are demonstrated by our LC-MS analysis of GS-MQ accumulation (Fig. 4B) and LMW content (Fig. 6) (the same samples were analyzed).

Moreover, does MRP1 up-regulation after knockdown rescue resistance to APR-246?

Author response: We thank the referee for raising this point regarding the association of MRP1 levels with APR-246 resistance. To address this question we have transiently transfected an MRP1 expression plasmid (Ito, Olsen et al., 2001) into HCT116 cells and treated the cells with APR-246. After 72h we determined cell viability by the WST-1 assay. Transfection of MRP1 did indeed reinstate resistance from APR-246, as shown in both HCT116 *TP53* WT and R248W *TP53* mutant cells. These results have been added to Fig. 1J-K and Appendix Fig. S1L-M (see below). In parallel we also transfected cells with an SLC7A11 construct as a positive control. These results are now included in Fig. S7D-E. In agreement with previously published data (Liu, Duong et al., 2017), SLC7A11 overexpression made mutant p53-expressing cells resistant to APR-246, while there was a very minor

change in sensitivity in cells harboring wild type p53. Overexpression of MRP1, on the other hand, decreased APR-246 sensitivity independently of *TP53* status.

Fig. 1J Western blot analysis of MRP1 (c.s.), p53 (DO-1) and GAPDH of HCT116 WT and R248W cells 72h after transfection with empty vector (EV) and MRP1 expression vector.

Fig. 1K Growth suppression (WST-1 assay) of HCT116 WT and R248W cells transfected with empty vector (EV) and MRP1 vector after 72h APR-246 treatment (n = 4). Data is also part of Fig S7E.

Fig. S1L Ponceau staining and size markers of western blot membrane in Fig. 1J.

Fig. S1M Effect of empty vector, MRP1 and xCT transfection on viability of HCT116 WT and HCT116 R248W cells without any drug treatment as determined by WST-1 assay. Data is fold change to average cell viability of empty vector transfected cells.

Fig. S7D Western blot analysis of xCT and GAPDH of HCT116 WT and R248W cells 72h after transfection with empty vector (EV) and xCT.

Fig. S7E Growth suppression (WST-1 assay) of HCT116 WT and R248W cells transfected with EV, MRP1 and xCT after 72h APR-246 treatment (n = 4). Data has partly been shown in Fig. 1K.

Extra Figure

Representative brightfield images of HCT116 *TP53* R248W cells transfected with empty vector (pcDNA3.1(-)) and MRP1 plasmid (MRP_k-pcDNA3.1(-)). EV transfected cells have many round visually dead cells after 72h treatment with 10 μ M APR-246 while MRP1 transfected cells do not.

Moreover (Fig. 1C) Among the "APR-246 high sensitivity" cell lines there is a p53 WT cell line A375 with a very low synergy index. Could you explain or make a hypothesis?

Author response: We have shown that APR-246 sensitivity is determined by several factors: presence of mutant p53, cellular thiol status and drug accumulation (Fig. 5). These are all factors that may affect synergy to APR-246 and MK-571. In terms of response to oxidative stress, melanoma differs from other cancer types as prooxidant pheomelanin production consumes cysteine (Chintala, Li et al., 2005), the GSH building block, and produces oxidative species (Denat, Kadarkar et al., 2014). On the other hand, eumelanin has antioxidative functions (Kim, Panzella et al., 2020). Krayem et al. have shown that APR-246 may be effective in wild type *TP53* melanoma cell lines (Krayem, Journe et al., 2016). Thus, a better understanding of the complex redox biology in association to melanin production may give clues as to why A375 has a low synergy index compared to e.g. EST140, another melanoma line with wild type *TP53*.

Referee #2 (Comments on Novelty/Model System for Author):

This work describes the effects of combined treatment with APR-246, a mutant p53 targeting drug, with inhibition of the efflux pump MRP1/ABCC1. The potential mechanism of action of APR-246 has been described extensively, involving MQ binding to cysteine residues. It has also been described that APR-246 (MQ) affects GSH and TXN antioxidant systems including the role of SLC7A11. Finally, an important role for MRP1/ABCC1 in redox regulation by controlling GSH/GSSG homeostasis has been described before. The work described here integrates this knowledge in the combined and enhanced

effects of inhibition of the different processes. From this work it is difficult to pinpoint the observed effects to a single mechanism as they strongly affect each other and all seem to contribute to the enhanced killing.

The model systems used are relevant and extent to in vivo analysis, the technical quality of the experiments is high and supports the conclusions in this manuscript.

With respect to the medical impact, the correlation with mutant p53 is not very strong and it seems that the drug combination is not specific for mutant p53 cancers. This could potentially limit the clinical use and impact.

Author response: We agree that the synergistic effect between APR-246 and MK-571 is not entirely restricted to *TP53* mutant tumors. APR-246 induces tumor cell death not only by reactivation of mutant p53 but also via induction of oxidative stress and depletion of cellular antioxidant capacity. It is widely accepted that cancer cells have decreased capacity to cope with oxidative stress compared to normal cells, and therefore show an increased susceptibility to electrophilic attack (Gorrini, Harris et al., 2013, Perillo, Di Donato et al., 2020). The latter mechanism can probably account for the observed effect of APR-246 on *TP53* WT or null tumors. This may in fact broaden the clinical use of APR-246 to tumors carrying WT *TP53*. Further clinical studies with APR-246 will hopefully provide more information to this question and identify relevant biomarkers.

Referee #2 (Remarks for Author):

The manuscript is well written, the experiments well performed and they strongly support the conclusions drawn from this work. As the authors state on page 4, lines 9-13, it remains difficult to separate the different parameters and therefore reach a strong conclusion on the contribution of the different cellular changes to the enhanced killing. However, the work present here is interesting and highlights the importance of redox homeostasis for the effects of APR-246.

Minor points for improvement:

1. Figure 3B, it would be informative to reflect the changes for each specific cell line rather than a gift in overall behavior.

Author response: We agree with the referee and have prepared a new figure that shows the changes in ^{14}C accumulation for individual cell lines (see below). An advantage of this figure is also that individual experiments and standard error of the mean are visible. The advantage of the initial figure is its visual simplicity and the statistical value from analyzing a larger group by a paired t test. Therefore, we have included the figure below as Figure EV3A. Furthermore, changes in ^{14}C accumulation for specific cell lines is also shown in Appendix table S3 which includes a column for fold change for each cell line and condition. As Fig. 6A is similar to Fig. 3B, we have modified this figure in a similar way and included it as Fig. EV6A.

Figure EV3

Fig. EV3A ^{14}C accumulation (cpm/mg x ml⁻¹) in 11 cell lines after 24h treatment with 10 or 25 μM ^{14}C -APR-246 +/- MK-571 (n \geq 3). Detailed information including exact n shown in Table S3.

Figure EV6

Fig. EV6A Total intracellular glutathione (GSH + GSSG) levels in six cell lines after 24h incubation +/- MK-571 as determined by a glutathione reductase (GR) re-cycling assay (n \geq 3 for each cell line except HCT116 -/- where n = 2, exact n shown by dot). Mean values for each cell line shown in Fig 6A.

2. In figure S4C, it seems that the equilibrium is more towards NAC-MQ while the reaction arrows suggest an equal distribution?

Author response: It is possible that the reaction is shifted towards NAC-MQ formation, but based on our data we cannot draw this conclusion. The left and right y-axes show arbitrary units (A.U.) for GS-MQ and NAC-MQ, respectively. Thus, one should compare the values quantitatively but rather look at the relative increase or decrease for each molecule. To clarify this point we have added the sentence “Values on y-axes are not comparable because GS-MQ and NAC-MQ have different response signals on MS.” to the figure legend highlighted in yellow. We have also changed the color of the values on the y-axes to further clarify that the green line corresponds to the left y-axis and purple line corresponds to the right y-axis.

Figure 4

D

Fig. 4D Amount of GS-MQ by LC-MS (green line, left axis) and NAC-MQ (purple line, right axis) over time after incubation of GS-MQ with NAC (n = 2). Values on y-axes are not comparable because GS-MQ and NAC-MQ have different response signals on MS.

3. The conclusions on page 7 line 17 and page 8 lines 11-12 seem to contradict each other. This should be corrected or more clearly explained.

Author response: We thank the referee for this pertinent point. We have changed these sentences on page 7 and modified the sentence on page 8 for a better clarity, as highlighted in yellow and copied below:

“Our results demonstrate that ¹⁴C-APR-246 accumulates inside cells upon MRP1 inhibition or siRNA knockdown, suggesting that either APR-246, MQ, MQ conjugates or all mentioned are exported via MRP1.”

“These data suggest that the inhibition of MRP1 does not lead to accumulation of prodrug APR-246 but results in retention of GS-MQ within the cell. This increases the cellular pool of active product MQ, which can bind to various intracellular thiols.”

4. The statement on page 8, lines 19-20 does not seem to be supported by Figure 1D-E, where mutant and null seem not significantly different.

Author response: : This statement refers specifically to the three isogenic cell systems HCT116, H1299 and Saos-2, not all tested cell lines. We have changed the figure references in the sentence on p. 8 as shown below.

“Mutant p53-expressing cells in all three isogenic cell systems tested (HCT116, H1299 and Saos-2), showed a more pronounced synergistic growth suppression compared to p53 null and WT cells (Fig. 1C and Fig. S1B)”

Referee #3 (Comments on Novelty/Model System for Author):

The main problem of the manuscript is the lack of robust in vivo experiments which are essential to validate the clinical value of the reported findings. The use of PDX of diverse types (at least two) has been indicated to the authors. Since APR-246 is already in clinical trials any additional co-treatment which should potentially reduced its toxicity and increase its efficacy needs to be tested as first step in in vivo experiments and only as a second step in cell lines. This main flaws makes the manuscript unsuitable for publication in EMBO MOL MED.

Referee #3 (Remarks for Author):

Wiman's group reports that pharmacological inhibition or siRNA-mediated depletion of MRP1 increases the anticancer activity of APR-246, a reactivator of mutant p53 proteins in human cancers. APR-246 clinical efficacy is actually tested in different clinical trials and the reported findings might represent the rationale for new therapeutic approaches of human cancer carrying TP53 mutations and expressing gain of function mutant p53 proteins.

While the pre-clinical and biochemical data are well designed and performed the lack of a broad range of in vivo experiments (for instance PDX derived from tumors carrying either TP53 mutations or intact TP53 gene) strongly flaws the overall message of the manuscript.

Indeed, the reported data provide additional evidence of the potential use of APR-246 in combination treatment but they do not provide any robust in vivo evidence that can support translation to clinical application.

Specific comments

1. Is there any correlation between TP53 status and expression of MRP1 in TCGA of different human cancers?

Author response: The potential correlation between *TP53* status and expression level of *ABCC1* (MRP1) mRNA was assessed using the TCGA dataset, and indeed there were some interesting findings. We have incorporated these findings in the revised Fig. 5G and Appendix Figure S5D (see below). We have also added comments relevant to the TCGA data on p. 11 in the Results section.

Fig. 5G ABCC1 mRNA Expression, RSEM (Batch normalized from Illumina HiSeq_RNASeqV2) in lung cancer (luad & lusc studies, * $p < 0.0001$) and colon adenocarcinoma (coad study, * $p = 0.005$) grouped into having no alterations or putative driver missense mutations in *TP53*. Statistical analysis by Mann Whitney test, n is indicated in the figure

Fig. S5 mRNA Expression, RSEM (Batch normalized from Illumina HiSeq_RNASeqV2) of ABCC1 in the TCGA PanCancer atlas of esophageal carcinoma (esca study) (mean and SEM are indicated, * $p = 0.01$) grouped into having no alterations in *TP53* or putative driver mutations (missense or truncating). Statistical analysis by Kruskal-Wallis test and Dunn's multiple comparisons test, n is indicated in the figure.

2. Pag.5 lines 22: "MRP1 protein expression did not show any obvious association to TP53 gene status, APR-246, IC50, or d-score (Fig. S1D)". This sentence refers to cell lines, but it seems to be in contrast with the rationale of the work that MRP1 could have a role in redox regulation and export of GSH-conjugated drugs increasing the resistance to PRIMA-1 only in a mutp53 cellular background (Fig. 1E).

Author response: MRP1 efflux activity is not only dependent on protein expression levels but also on cellular levels of GSH and ATP. Therefore, MRP1 protein levels do not necessarily reflect MRP1 efflux pump activity. We have modified the Results section, p. 6, as shown below.

“In our relatively small panel of tested cell lines MRP1 protein expression did not show any obvious association to TP53 gene status, APR-246 IC50, or δ -score (Fig. S1D). However, as MRP1 efflux activity is GSH- and ATP-dependent (Hooijberg, Pinedo et al., 2000), MRP1 protein expression may not necessarily reflect its efflux pump activity.”

3. Protein expression analysis upon MK-571 treatment needs to be performed in different cell lines and not only for OVCAR-3.

Author response: We have performed Western blot analysis as in Fig. 6D in two more cell lines, HCT116 R248W and HCT116 WT *TP53*, and added the results in the revised Fig. 6 as shown below. Western blot analysis of xCT protein expression after APR-246 and MK-571 treatment in TOV-11D (R175H) and KADA (R248W) cells is also shown in Fig. S6E-F. Western blot analysis of xCT expression after treatment with MK-571 alone in H1299 and HCT116 cells with different *TP53* status is shown in Fig. S6G.

Figure 6

Fig. 6E Western blot of HCT 116 WT and R248W cells treated with APR-246 +/- MK-571 for 24h.

4. The only set of in vivo experiments was performed by using Eso26 cells (no IC50 was identified). What about synergy with APR-246? This cell lines was not characterized along the manuscript. These experiments need to be performed by using different cell lines as some of those analyzed in Fig. 1.

Author response: Growth suppression data of the esophageal cell lines have been moved from Fig. S2 from the initially submitted version of the manuscript to the current version of Fig. EV1 for better clarity and transition to Fig. 2. The IC50 values for APR-246 +/- MK-571 for all esophageal cell lines are now included in Fig. 1C (previously Fig. 1D) and Appendix Table S1. Also, we have re-analyzed synergy in all cell lines in order to include the esophageal cell lines in this analysis in the revised Fig. 1D (previously Fig. 1E). We have also added a column in the Appendix table S1 with synergy scores including those of the esophageal cell lines (as shown below).

Figure 1

Fig. 1C Mean IC₅₀ values (µM) for APR-246 +/- 20 µM MK-571 of 21 cell lines. *p < 0.0001, Wilcoxon test. Mean IC₅₀ values and n for individual cell lines are shown in Table S1.

Fig. 1D ZIP synergy scores of most synergistic area sub-grouped according to TP53 gene status for 21 cell lines. Scores and n are shown in Table S1.

Table S1. IC₅₀ values of APR-246 with and without MK-571 and synergy scores, related to Figure 1, S1 and S2

Assay (conc. range to determine IC ₅₀ and synergy)	Cell line (TP53 status)	n (*)	IC ₅₀ (µM)			Most synergistic area score (up to 20µM MK-571)		
			APR-246	APR-246 + 10 µM MK-571	APR-246 + 20 µM MK-571	ZIP	Bliss	HSA
WST (0-15 or 20µM)	A375 (WT)	5 (4)	10	13	11	3	3	-2
	EST-140 (WT)	3 (1)	≥ 30	15	5	23	25	27
	EST-037 (C229fs)	3	≥ 30	12	6	32	30	32
	EST-049 (C275W)	3	10	6	3	16	17	18
	H1299 (-/-)	7 (6)	≥ 30	18	11	22	22	21
	H1299 (R175H)	7 (5)	16	8	5	24	24	30
	HCT 116 (-/-)	7 (3)	20	12	7	33	32	34
	HCT 116 (R248W)	5 (3)	14	7	4	36	36	37
	HCT 116 (WT)	6 (3)	22	13	8	24	23	27
	HDF (WT)	3	≥ 30	26	17	4	4	3
	KADA (R248W)	4	18	9	8	29	29	30
	LNCaP (WT)	3	≥ 30	23	14	22	21	22
	OVCAR-3 (R248Q)	6 (3)	12	8	3	46	46	50
	Saos-2 (-/-)	5	≥ 30	15	10	11	11	7
	Saos-2 (R273H)	5 (4)	27	9	7	33	33	38
	SKMEL-2 (G245S)	3	6	3	3	17	18	14
TOV-112D (R175H)	3	≥ 30	13	8	34	34	39	
Resazurin (0-100µM)	Eso26 (R248W)	3	31	22	16	16	18	22
	FLO-1 (C277F)	3	35	35	34	5	5	4
	JH-EsoAd1 (G266E)	3	38	27	20	35	34	35
	OACM5.1 (R248Q)	3	30	25	17	29	27	29

* number of experiments for combination treatment with 20 µM MK-571 when n is different for this combination treatment.

5. IHC for p53, MRP1 and ki67 expression needs to be performed in the xenografted tumors.

Author response: We have performed p53, MRP1 and Ki67 immunostaining and included quantification and representative images in revised Figs. 2 and S2 at two different time points. We have also stained for cleaved caspase 3 as an indicator for apoptosis. Ki67 is decreased and cleaved caspase 3 is increased upon combination treatment at the later timepoints, reflecting the change in tumor volume. The Eso26 xenograft expresses high levels of p53, in accordance with the presence of a hot spot missense TP53 mutation. Eso26 also expresses MRP1 as shown by the immunostaining.

6. Figure 6 shows the shifts cellular thiol pools by MRP1 inhibitor MK-571 seem to be rather modest.

Author response: The thiol shifts shown in Fig.6 may seem modest but we believe that the intracellular consequences of these changes could be rather substantial. Cysteine is the most chemically reactive amino acid found in cells due to its thiol group (SH), and it is mainly found in proteins or in GSH (Hansen, Roth et al., 2009). Several publications show that glutathione is decreased upon treatment with MK-571 (Cullen, Davey et al., 2001, Minich, Riemer et al., 2006, Stefan & Wiese, 2019, Wen, Iwata et al., 2019, Whitt, Keeton et al., 2016). In agreement with these studies, our data in Fig. 6A and EV6A show a robust decrease in total glutathione (GSH + GSSG) in five out of six cancer cell lines after MK-571 treatment. On average, total glutathione (GSH+GSSG) is decreased by 54% in these five cell lines, a drop to approximately half. This would have a significant impact on the antioxidant defense and render cells more sensitive to electrophiles. Glutathione is the most predominant intracellular LMW thiol (mM range) while cysteine is the most predominant thiol extracellularly (Giustarini, Dalle-Donne et al., 2009). In response to the drop in glutathione we observed an increased expression of xCT/SLC7A11 and an intracellular accumulation of cysteine. As imported cystine is highly insoluble and can cause disulfide toxicity (Joly, Delfarah et al., 2020, Liu, Olszewski et al., 2020), cystine is efficiently reduced to cysteine, which may explain why we see a more than 2.3-fold increase in cysteine (Fig. 6G) and only a modest 37% increase in its oxidized form

(CySS) after treatment of MK-571 compared to untreated cells (Fig. 6F). We have changed the scale on the y-axis in Fig. 6F to make this difference more visible.

Figure 6

Fig. 6F Intracellular cystine (CySS per 10^6 cells) as shown by LC-MS in OVCAR-3 cells at 24h treatment with APR-246 +/- MK-571 (n = 3).

Dear Dr. Wiman,

Thank you for your note asking whether for your revised article, the use of PDOs could be an alternative to PDXs. I have now contacted referee #3 who was insisting on PDX models and this referee would be perfectly satisfied with PDOs. This referee added the following: "Patients derived organoids are fine for me. They represent a very good in vivo tool. As you pointed out number of patients is important, but more important, they [PDOs] need to be representative of at least two types of tumours carrying hotspot p53 mutations."

I hope this is helpful.

I am looking forward to receiving the revised article when ready.

With my best wishes,

Celine

Celine Carret, PhD
Senior Editor
EMBO Molecular Medicine
EMBO Press

Author response: We have performed experiments with two collaborating groups to examine the effect of combination treatment with APR-246 and MK-571 on patient-derived organoids (PDOs) representing two tumor types relevant to our study, colorectal cancer and esophageal cancer. The three esophageal cancer PDOs carry *TP53* hot spot mutations R248W, R175H and R248Q, while the two colorectal cancer PDOs have *TP53* missense mutations H214R and E224D. PDO is a translational preclinical model based on 3D culture that preserves tumor heterogeneity and is considered to predict clinical efficacy of a drug or drug combination with high accuracy (Li, Francies et al., 2018, Paquet-Fifield, Koh et al., 2018).

For the first time we demonstrate cytotoxic efficacy of APR-246 in PDOs derived from five different patients. We also show that MK-571 can lower the IC50 of APR-246 in the tested PDOs. We observed variation in the results between different experiments, presumably due to tumor heterogeneity.

However, most experiments demonstrated synergy as determined by the ZIP model based on CTG assay (Fig. 2I). The colorectal cancer PDOs showed more potent synergy which is in accordance with the *in vitro* experiments in cell lines (Table S1). Beside CTG assay we also did an image analysis that assessed the organoid area and PI intensity. Fig. 2H convincingly shows an increase in PI intensity upon combination treatment and thus synergistic cell death in colo-PDOs. We have included the new PDO data in Fig. 2 and added a Table S2 for detailed information regarding the PDOs.

Figure 2

Figure EV2

Figure S2

Fig 2F Growth suppression determined by the ATP-based CTG assay in colorectal cancer patient-derived organoids (colo-PDO) after treatment with APR-246 and 20 μ M MK-571 in colo-PDO1 and 40 μ M MK-571 in colo-PDO2 as these were less sensitive to MK-571. n = 3.

Fig 2G IC50 values (μ M) for APR-246 +/- 20 μ M MK-571 in indicated PDOs. *p = 0.04, paired t test. Each dot indicates n for individual PDOs.

Fig 2H Organoid area (in black, left y-axis) and PI intensity (in red, right y-axis) of colo-PDO1 and colo-PDO2 as determined by image analysis 72h after treatment with APR-246 +/- MK-571. n = 3.

Fig 2I Highest synergy score according to the ZIP model based on growth suppression as shown by CTG assay (ATP-based) or image analysis (Area and PI) in eso-PDOs and colo-PDOs. Scores are shown on a log scale. Score above 0 indicates synergy. Each dot indicates n for individual PDOs.

Fig EV2A

References:

- Chintala S, Li W, Lamoreux ML, Ito S, Wakamatsu K, Sviderskaya EV, Bennett DC, Park YM, Gahl WA, Huizing M, Spritz RA, Ben S, Novak EK, Tan J, Swank RT (2005) Slc7a11 gene controls production of pheomelanin pigment and proliferation of cultured cells. *Proceedings of the National Academy of Sciences of the United States of America* 102: 10964-9
- Cullen KV, Davey RA, Davey MW (2001) Verapamil-stimulated glutathione transport by the multidrug resistance-associated protein (MRP1) in leukaemia cells. *Biochemical pharmacology* 62: 417-24
- Denat L, Kadekaro AL, Marrot L, Leachman SA, Abdel-Malek ZA (2014) Melanocytes as instigators and victims of oxidative stress. *J Invest Dermatol* 134: 1512-1518
- Gana CC, Hanssen KM, Yu DMT, Flemming CL, Wheatley MS, Conseil G, Cole SPC, Norris MD, Haber M, Fletcher JI (2019) MRP1 modulators synergize with buthionine sulfoximine to exploit collateral sensitivity and selectively kill MRP1-expressing cancer cells. *Biochemical pharmacology* 168: 237-248
- Giustarini D, Dalle-Donne I, Milzani A, Rossi R (2009) Oxidative stress induces a reversible flux of cysteine from tissues to blood in vivo in the rat. *FEBS J* 276: 4946-58
- Gorrini C, Harris IS, Mak TW (2013) Modulation of oxidative stress as an anticancer strategy. *Nature reviews Drug discovery* 12: 931-47
- Haffo L, Lu J, Bykov VJN, Martin SS, Ren X, Coppo L, Wiman KG, Holmgren A (2018) Inhibition of the glutaredoxin and thioredoxin systems and ribonucleotide reductase by mutant p53-targeting compound APR-246. *Sci Rep* 8: 12671. doi: 10.1038/s41598-018-31048-7.
- Hansen RE, Roth D, Winther JR (2009) Quantifying the global cellular thiol-disulfide status. *Proceedings of the National Academy of Sciences of the United States of America* 106: 422-7
- Hipfner DR, Almquist KC, Leslie EM, Gerlach JH, Grant CE, Deeley RG, Cole SP (1997) Membrane topology of the multidrug resistance protein (MRP). A study of glycosylation-site mutants reveals an extracytosolic NH2 terminus. *The Journal of biological chemistry* 272: 23623-30
- Ito K, Olsen SL, Qiu W, Deeley RG, Cole SP (2001) Mutation of a single conserved tryptophan in multidrug resistance protein 1 (MRP1/ABCC1) results in loss of drug resistance and selective loss of organic anion transport. *The Journal of biological chemistry* 276: 15616-24
- Joly JH, Delfarah A, Phung PS, Parrish S, Graham NA (2020) A synthetic lethal drug combination mimics glucose deprivation-induced cancer cell death in the presence of glucose. *The Journal of biological chemistry* 295: 1350-1365
- Kim E, Panzella L, Napolitano A, Payne GF (2020) Redox Activities of Melanins Investigated by Electrochemical Reverse Engineering: Implications for their Roles in Oxidative Stress. *J Invest Dermatol* 140: 537-543

Krayem M, Journe F, Wiedig M, Morandini R, Najem A, Sales F, van Kempen LC, Sibille C, Awada A, Marine JC, Ghanem G (2016) p53 Reactivation by PRIMA-1(Met) (APR-246) sensitises (V600E/K)BRAF melanoma to vemurafenib. *Eur J Cancer* 55:98-110.: 10.1016/j.ejca.2015.12.002. Epub 2016 Jan 17.

Lambert JM, Gorzov P, Veprintsev DB, Soderqvist M, Segerback D, Bergman J, Fersht AR, Hainaut P, Wiman KG, Bykov VJ (2009) PRIMA-1 reactivates mutant p53 by covalent binding to the core domain. *Cancer cell* 15: 376-88

Li X, Francies HE, Secrier M, Perner J, Miremadi A, Galeano-Dalmau N, Barendt WJ, Letchford L, Leyden GM, Goffin EK, Barthorpe A, Lightfoot H, Chen E, Gilbert J, Noorani A, Devonshire G, Bower L, Grantham A, MacRae S, Grehan N et al. (2018) Organoid cultures recapitulate esophageal adenocarcinoma heterogeneity providing a model for clonality studies and precision therapeutics. *Nature communications* 9: 2983

Liu DS, Duong CP, Haupt S, Montgomery KG, House CM, Azar WJ, Pearson HB, Fisher OM, Read M, Guerra GR, Haupt Y, Cullinane C, Wiman KG, Abrahmsen L, Phillips WA, Clemons NJ (2017) Inhibiting the system xC⁻/glutathione axis selectively targets cancers with mutant-p53 accumulation. *Nature communications* 8: 14844

Liu X, Olszewski K, Zhang Y, Lim EW, Shi J, Zhang X, Zhang J, Lee H, Koppula P, Lei G, Zhuang L, You MJ, Fang B, Li W, Metallo CM, Poyurovsky MV, Gan B (2020) Cystine transporter regulation of pentose phosphate pathway dependency and disulfide stress exposes a targetable metabolic vulnerability in cancer. *Nature cell biology* 22: 476-486

Minich T, Riemer J, Schulz JB, Wielinga P, Wijnholds J, Dringen R (2006) The multidrug resistance protein 1 (Mrp1), but not Mrp5, mediates export of glutathione and glutathione disulfide from brain astrocytes. *Journal of neurochemistry* 97: 373-84

Paquet-Fifield S, Koh SL, Cheng L, Beyit LM, Shembrey C, Molck C, Behrenbruch C, Papin M, Gironella M, Guelfi S, Nasr R, Grillet F, Prudhomme M, Bourgaux JF, Castells A, Pascussi JM, Heriot AG, Puisieux A, Davis MJ, Pannequin J et al. (2018) Tight Junction Protein Claudin-2 Promotes Self-Renewal of Human Colorectal Cancer Stem-like Cells. *Cancer research* 78: 2925-2938

Perillo B, Di Donato M, Pezone A, Di Zazzo E, Giovannelli P, Galasso G, Castoria G, Migliaccio A (2020) ROS in cancer therapy: the bright side of the moon. *Exp Mol Med* 52: 192-203

Pillai-Kastoori L, Heaton S, Shiflett SD, Roberts AC, Solache A, Schutz-Geschwender AR (2020) Antibody validation for Western blot: By the user, for the user. *The Journal of biological chemistry* 295: 926-939

Shukalek CB, Swanlund DP, Rousseau RK, Weigl KE, Marensi V, Cole SP, Leslie EM (2016) Arsenic Trigluthathione [As(GS)₃] Transport by Multidrug Resistance Protein 1 (MRP1/ABCC1) Is Selectively Modified by Phosphorylation of Tyr920/Ser921 and Glycosylation of Asn19/Asn23. *Molecular pharmacology* 90: 127-39

Stefan SM, Wiese M (2019) Small-molecule inhibitors of multidrug resistance-associated protein 1 and related processes: A historic approach and recent advances. *Med Res Rev* 39: 176-264. doi: 10.1002/med.21510. Epub 2018 May 29.

Stride BD, Grant CE, Loe DW, Hipfner DR, Cole SP, Deeley RG (1997) Pharmacological characterization of the murine and human orthologs of multidrug-resistance protein in transfected human embryonic kidney cells. *Molecular pharmacology* 52: 344-53

Wen X, Iwata K, Ikuta K, Zhang X, Zhu K, Ibi M, Matsumoto M, Asaoka N, Liu J, Katsuyama M, Yabe-Nishimura C (2019) NOX1/NADPH oxidase regulates the expression of multidrug resistance-associated protein 1 and maintains intracellular glutathione levels. *Febs J* 17: 14753

Whitt JD, Keeton AB, Gary BD, Sklar LA, Sodani K, Chen ZS, Piazza GA (2016) Sulindac sulfide selectively increases sensitivity of ABCC1 expressing tumor cells to doxorubicin and glutathione depletion. *J Biomed Res* 30: 120-133. doi: 10.7555/JBR.30.20150108. Epub 2015 Nov 20.

3rd Nov 2020

Dear Prof. Wiman,

Thank you for the submission of your revised manuscript to EMBO Molecular Medicine. We have now received the enclosed reports from the 2 referees who re-reviewed your manuscript. As you will see, they are supportive of publication, and I am thus pleased to inform you that we will be able to accept your manuscript pending the following final minor amendments:

1) Main manuscript text:

- Please answer/correct the changes suggested by our data editors in the main manuscript file (in track changes mode). This file will be sent to you in the next couple of days. Please use this file for any further modification.
- Please remove the yellow highlighted and the strikethrough text.
- Material and methods: Please include the full statement that the experiments conformed to the principles set out in the WMA Declaration of Helsinki and the Department of Health and Human Services Belmont Report (this sentence should also be included in the checklist).
- Data availability section: Primary datasets produced in this study need to be deposited in an appropriate public database (see <https://www.embopress.org/page/journal/17574684/authorguide#dataavailability>). The accession numbers and database should be listed in a formal "Data Availability" section (placed after Materials & Method). Please note that the Data Availability Section is restricted to new primary data that are part of this study. When not applicable, please indicate: "This study includes no data deposited in external repositories".
- References: Please update the format of the references so as to have 10 authors listed before "et. al".

2) Figures: Your EV figures are numbered Fig. EV1 to EV6, but Fig. EV4 is missing. Please update (figures, main manuscript text and legends).

3) We would also encourage you to include the source data for figure panels that show essential data. Numerical data should be provided as individual .xls or .csv files (including a tab describing the data). For blots or microscopy, uncropped images should be submitted (using a zip archive if multiple images need to be supplied for one panel). Additional information on source data and instruction on how to label the files are available at

In particular, please provide source data for Figure S5.

4) Appendix: We note that you have included material and methods in the appendix. You are welcome to include them in the main manuscript text instead.

5) For more information: There is space at the end of each article to list relevant web links for further consultation by our readers. Could you identify some relevant ones and provide such information as well? Some examples are patient associations, relevant databases, OMIM/proteins/genes links, author's websites, etc...

6) As part of the EMBO Publications transparent editorial process initiative (see our Editorial at <http://embomolmed.embopress.org/content/2/9/329>), EMBO Molecular Medicine will publish online a

Review Process File (RPF) to accompany accepted manuscripts.

In the event of acceptance, this file will be published in conjunction with your paper and will include the anonymous referee reports, your point-by-point response and all pertinent correspondence relating to the manuscript. Let us know whether you agree with the publication of the RPF and as here, if you want to remove or not any figures from it prior to publication.

I look forward to receiving your revised manuscript.

Yours sincerely,

Lise Roth

Lise Roth, PhD
Editor
EMBO Molecular Medicine

To submit your manuscript, please follow this link:

Link Not Available

Photos 400-800 DPI

*Additional important information regarding figures and illustrations can be found at <https://bit.ly/EMBOPressFigurePreparationGuideline>

The system will prompt you to fill in your funding and payment information. This will allow Wiley to send you a quote for the article processing charge (APC) in case of acceptance. This quote takes into account any reduction or fee waivers that you may be eligible for. Authors do not need to pay any fees before their manuscript is accepted and transferred to our publisher.

***** Reviewer's comments *****

Referee #1 (Remarks for Author):

The authors provided satisfactory results and comments.

Referee #3 (Comments on Novelty/Model System for Author):

I have carefully looked at the new set of data on PDOs derived from esophageal and colon cancers. All of them carry hotspot TP53 mutations. PDOs have been investigated using diverse experimental assays. The results are quite convincing (see new Fig.2 and related supplementary). Overall the translational value of the manuscript has been significantly increased. In its present form the manuscript is suitable for publication in EMBO Molecular Medicine.

The authors performed the requested changes.

17th Nov 2020

Dear Prof. Wiman,

We are pleased to inform you that your manuscript is accepted for publication and is now being sent to our publisher to be included in the next available issue of EMBO Molecular Medicine!

We would like to remind you that as part of the EMBO Publications transparent editorial process initiative, EMBO Molecular Medicine will publish a Review Process File online to accompany accepted manuscripts. If you do NOT want the file to be published or would like to exclude figures, please immediately inform the editorial office via e-mail.

Please read below for additional IMPORTANT information regarding your article, its publication and the production process.

Congratulations on your interesting work,

Sincerely,

Lise Roth

Lise Roth, Ph.D
Scientific Editor
EMBO Molecular Medicine

Follow us on Twitter @EmboMolMed
Sign up for eTOCs at embopress.org/alertsfeeds

*** ** IMPORTANT INFORMATION ** **

SPEED OF PUBLICATION

The journal aims for rapid publication of papers, using the advance online publication "Early View" to expedite the process: A properly copy-edited and formatted version will be published as "Early View" after the proofs have been corrected. Please help the Editors and publisher avoid delays by providing e-mail address(es), telephone and fax numbers at which author(s) can be contacted.

Should you be planning a Press Release on your article, please get in contact with embomolmed@wiley.com as early as possible, in order to coordinate publication and release dates.

LICENSE AND PAYMENT:

All articles published in EMBO Molecular Medicine are fully open access: immediately and freely available to read, download and share.

EMBO Molecular Medicine charges an article processing charge (APC) to cover the publication costs. You, as the corresponding author for this manuscript, should have already received a quote with the article processing fee separately. Please let us know in case this quote has not been received.

Once your article is at Wiley for editorial production you will receive an email from Wiley's Author Services system, which will ask you to log in and will present you with the publication license form for completion. Within the same system the publication fee can be paid by credit card, an invoice, pro forma invoice or purchase order can be requested.

Payment of the publication charge and the signed Open Access Agreement form must be received before the article can be published online.

PROOFS

You will receive the proofs by e-mail approximately 2 weeks after all relevant files have been sent to our Production Office. Please return them within 48 hours and if there should be any problems, please contact the production office at embopressproduction@wiley.com.

Please inform us if there is likely to be any difficulty in reaching you at the above address at that time. Failure to meet our deadlines may result in a delay of publication.

All further communications concerning your paper proofs should quote reference number EMM-2019-10852-V4 and be directed to the production office at embopressproduction@wiley.com.

Thank you,

Lise Roth, Ph.D
Scientific Editor
EMBO Molecular Medicine

Corresponding Author Name: Klas Wiman
Journal Submitted to: EMBO Molecular Medicine
Manuscript Number: EMM-2019-10852-V2-Q